# SiT: Symmetry-invariant Transformers for Generalisation in Reinforcement Learning

## Abstract

An open challenge in reinforcement learning (RL) is the effective deployment of a trained policy to new or slightly different situations, i.e. out-of-distribution data, as well as semantically-similar environments. To overcome these limitations, we introduce **S**ymmetry-**I**nvariant **T**ransformer (SiT), a scalable vision transformer (ViT) that inherently identifies and leverages both local and global data patterns in a self-supervised manner. Central to our approach is Graph Symmetric Attention, which refines the traditional self-attention process to preserve graph symmetries, resulting in invariant and equivariant latent representations. SiT adeptly balances the recognition of local patterns with broader data trends, allowing it generalize to unseen data distributions. We showcase SiT's superior generalization over ViTs on MiniGrid and Procgen RL benchmarks, and its sample efficiency on Atari 100k.

## 1 Introduction

Despite recent advances in reinforcement learning, out-of-distribution generalization remains an open challenge. A widely-used approach to improve generalisation in image-based RL is data augmentation (Laskin et al., 2020; Yarats et al., 2021b; Hansen & Wang, 2021) but it can lead to over regularisation to specific augmentations. Moreover, data augmentation's inherent non-determinism can amplify the variance in regression targets, which can be detrimental to learning (Hansen et al., 2021). Complimentary to data augmentation, leveraging symmetries can improve generalization and lead to sample-efficient RL (Tang & Ha, 2021; Van der Pol et al., 2020; Weissenbacher et al., 2022).

Image-based RL may benefit from both local and global symmetries, which preserve a particular structure or property within a neighborhood of a pixel or image patches and throughout the entire image respectively. However, enforcing local symmetries through data augmentation can be sample inefficient and computationally expensive. This is because when an image is divided into local patches to capture these symmetries, the number of augmented samples we may need to represent all possible variations grows exponentially. Given the prevalence of symmetries in RL settings, it is advantageous for neural networks to possess the capability to develop a understanding of these local and global symmetries in a self-supervised manner that is data-driven.

However, leveraging symmetries in RL presents various challenges. In particular, an agent's action choices in general are not invariant under symmetries both globally and locally, see Figure 1. Permutation invariance (Tang & Ha, 2021) in Figure 1 (a) admits the shortcoming that it leads to dead-end situations in many settings, while local and global flip symmetries (b) inter-changes left /right and up / down actions. Moreover, in many scenarios, it's essential for a decision-making process to consider the local context within the broader global setting, e.g., in Figure 1 (c), the global 90° rotation is an exact symmetry but local patch-wise rotations change the neighbourhood of the agent. In contrast, even minimal permutations are fatal for learning, see Figure 1 (a) bottom-right. This situation is common amongst many games and real-world environments (Bellemare et al., 2012; Cobbe et al., 2020; Silver et al., 2016; Bellemare et al., 2020; Kitano et al., 1997).

To address the aforementioned challenges, we present a self-attention based network architecture, which we call **S**ymmetry-**i**nvariant **T**ransformer (SiT). SiTs incorporates a flexible relational inductive bias (Battaglia et al., 2018) to recognize relational patterns or symmetries, enabling it to adapt effectively to unfamiliar or out-of-distribution data. In addition to invariance, SiTs account for dead-end situations by incorporating equivariance, which refers to the property of an action to transform equivalently as the states under symmetries in our SiT module and by introducing a rotation

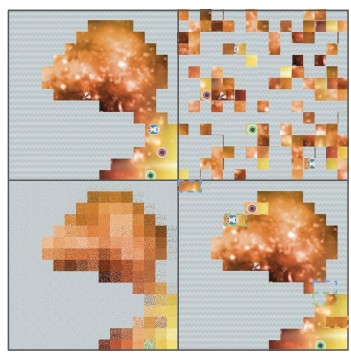
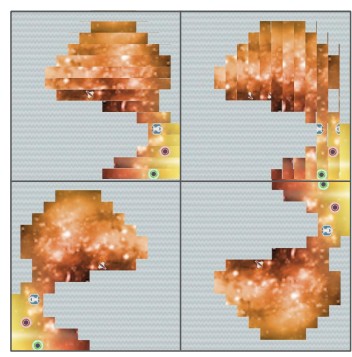
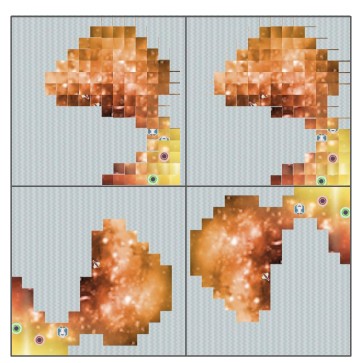
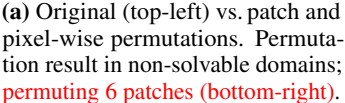

**(a)** Original (top-left) vs. patch and pixel-wise permutations. Permutation result in non-solvable domains; permuting 6 patches (bottom-right).

**(b)** Horizontal and vertical flips; patch-wise (top) - entire image (bottom). Approximate symmetry as it breaks left / right actions.

**(c)** Left and right rotations. Patchwise (top): agent's local neighbourhood altered; Entire image: exact symmetry (bottom).

**Figure 1:** Local (patch-wise) and global transformations of observations of the CaveFlyer environment, Procgen suite (Cobbe et al., 2020). Permutation invariant agents Tang & Ha (2021) can't discern key features (a) in contrast to agents with local and/or global flip and rotation invariance (b) and (c).

symmetry preserving but flip-symmetry breaking layer. Additionally, we introduce novel invariant as well as equivariant **G**raph **S**ymmetric **A**ttention (GSA). GSA is akin to self-attention of Vision Transformers (ViTs) (Dosovitskiy et al., 2020), by adapting permutation-invariant self-attention (Lee et al., 2019) to maintain graph symmetries.

SiTs capitalize on the interplay between local and global information. This is achieved by incorporating both local and global GSA modules. In particular, the local attention window stretches over several image patches, such that the local symmetries do not change the agent's local broader neighbourhood. We demonstrate the efficacy of SiTs over ViTs on prevalent RL generalization benchmarks, namely MiniGrid and Procgen, and the CIFAR-10 vision benchmark.

In summary, our contribution is threefold. First, we introduce a scalable invariant as well as equivariant transformer architecture (SiT), i.e. accounting for symmetries down to the pixel level (Section 4). In contrast to ViTs, SiTs require less hyper-parameter tuning and generalise better in RL tasks. Specifically, SiTs lead to a $3\times$ and $9\times$ improvement in performance over ViTs on commonly-used MiniGrid and Procgen environments. Secondly, SiTs incorporate a novel method to account for the interplay of local and global symmetries, which is complementary to widely-used data augmentation in image-based RL. Third, GSA is a novel approach to accomplish graph-symmetries in self-attention, not relying on positional embeddings (Fuchs et al., 2020; Romero & Cordonnier, 2021).

## 2 BACKGROUND

**Reinforcement Learning.** A Markov Decision Process (MDP) is a mathematical framework for modeling decision-making problems in stochastic environments. MDPs are characterized by a tuple $(\mathcal{S}, \mathcal{A}, \mathcal{P}, \mathcal{R}, \gamma)$, where $\mathcal{S}$ is a finite set of states, $\mathcal{A}$ is a finite set of actions, $\mathcal{P}$ is the transition probability function, $R$ is the reward function, and $\gamma \in [0, 1)$ is the discount factor. In RL, one aims to learn optimal decision-making policies in MDPs. A policy, denoted as $\pi : \mathcal{S} \to \mathcal{A}$, is a mapping from states to actions. The optimal policy $\pi$ maximizes the expected cumulative discounted reward, given by the value function $V^\pi(s) = \mathbb{E}\left[\sum_{t=0}^\infty \gamma^t \mathcal{R}(s_t, a_t) \mid s_0 = s, a_t \sim \pi(\cdot \mid s_t)\right]$, where the expectation is taken over the sequence of states and actions encountered by following the policy $\pi$. The optimal policy $\pi^*$ is the one that satisfies $V^{\pi^*}(s) \geq V^\pi(s)$ for all $s \in \mathcal{S}$ and any other policy $\pi$.

**Invariance and Equivariance**. Invariance and equivariance are foundational concepts in understanding how functions respond to symmetries of their inputs. In RL, equivariance and invariance properties are imposed on the actor and value networks, e.g. in Wang et al. (2023) on top of SAC (Haarnoja et al., 2018). Before defining these concepts, we introduce some notation. The function $f$ maps elements from space $\mathcal{S}$ to $\mathcal{S}'$ and $g$ denotes an individual transformation in a symmetry group. The functions $\rho(g)$ and $\rho'(g)$ describe the action of g on spaces $\mathcal{S}$ and $\mathcal{S}'$, respectively, e.g. $\rho(g) \cdot s$ signifies applying a transformation $\rho(g)$ on an element $s$ of $\mathcal{S}$.

- Invariance: A function $f$ is invariant with respect to a set of transformations (symmetry group) if the application of any transformation from this set to its input does not change the function's output. Mathematically, this is expressed as: $f(\rho(g) \cdot s) = f(s)$ for every transformation $g$.

- Equivariance: A function $f$ is equivariant if, when a transformation is applied to its input, there is a corresponding and predictable transformation of its output. This relationship is captured by the equation: $f(\rho(g) \cdot s) = \rho'(g) \cdot f(s)$ for every transformation $g$ in the symmetry group.

**Attention mechanisms.** Recently, conventional self-attention have been employed in the context of RL agents (Tang & Ha, 2021). The permutation invariant self-attention layer (Lee et al., 2019) uses a fixed Q-matrix (queries). The original ViT architecture (Dosovitskiy et al., 2020) naturally admits permutation invariance (PI) due to use of token embeddings. PI is only broken by using the positional embedding (Romero & Cordonnier, 2021; Fuchs et al., 2020). The standard attention is given by

$$\text{Att}(K, V, Q) \quad = \quad \text{softmax}\left( \tfrac{1}{\sqrt{d_f}} Q\,K^T \right) V \ , \tag{1}$$

where $K, V,$ and $Q$ denote the keys, values, and queries respectively. They are derived from the input $X$: $K = XW^k$, $V = XW^v$, $Q = XW^q$, where $W^q$, $W^k$, and $W^v$ are the corresponding weight matrices. The first component of $K, V, Q$ inherits the token embedding. The keys and values are constructed based on the input data, which is segmented into $P$ patches. Consequently, the matrices $K, V,$ and $Q$ have dimensions $\mathbb{R}^{P \times d_f}$, where $d_f$ represents the feature dimension for each patch.

Graph neural networks and graph attention have been extensively explored in terms of their symmetries (Veličković et al., 2018; Satorras et al., 2021b). At a high-level, the graph attention mechanism (GAT) determines the relationships between nodes in a graph using attention. In essence, it constructs an attention matrix, derived from the score matrix $\Gamma$, which is masked with the adjacency matrix $\mathcal{G}$ to ensure that the attention coefficients are only computed for nodes that are connected in the graph

$$\text{GAT}\,(K, V, Q) \quad = \quad \text{softmax}\left( \tfrac{1}{\sqrt{d_f}} \Gamma(Q, K) \right) \mathcal{G}\,V, \quad \text{with } \Gamma(Q, K) = QK^T \ , \tag{2}$$

where $K, V, Q$ are the feature vectors of the nodes, multiplied with weight matrices. The softmax translates $\Gamma(V)$ into probabilities, emphasizing certain node connections. Optionally a symmetrisation of the score matrix may be added.[1] The latter, ensures that connections between nodes are bidirectional, meaning their importance is consistent regardless of direction.

## 3 GSA: Symmetry-Invariant and Equivariant Attention

In this work, we propose a modification of the permutation invariant attention layer (Lee et al., 2019). This adaptation is specifically designed to respect the inherent symmetries of a square two-dimensional grid, which serves as our underlying graph structure. These symmetries include translations, rotations, and flips, as depicted in Figure 2. Our approach is an evolution of the rotary embedding method (Su et al., 2021). Our **G**raph **S**ymmetric **A**ttention (GSA) layer is conceptually similar to a traditional graph-adjacency matrix. Our graph topology matrix G is the analog of the adjacency matrix in equation 2; however, its trainable weights are uniquely constrained to abide by certain symmetry conditions, which we discuss later. While our discussion centers on the 2D grid, GSA may be adapted to 1D data where it ensures shift-symmetry (optionally flip-symmetry), see A.1.

For clarity, imagine a $9 \times 9$ pixel image. When segmented into $3 \times 3$ pixel patches, we get 9 distinct patches. In the **local** GSA setup, each graph vertex corresponds to an individual pixel, suggesting that in Figure 2, the term "patches" is synonymous with pixels. In contrast, the **global** GSA interprets the image as a collection of $3 \times 3$ patches, where each patch's central point is symbolized by a graph vertex, aligning with the conventional Vision Transformer perspective.

Taking inspiration from self-attention in graphs, we propose **G**raph **S**ymmetric **A**ttention (GSA):

$$\text{GSA}(K, V, Q) \quad = \quad \text{softmax}\left( \tfrac{1}{\sqrt{d_f}} \Gamma(Q, K) \right) G_v\,V \tag{3}$$

$$\text{with } \Gamma(Q, K) \quad = \quad \text{symmetric}\left( \left( G_q\,Q\,[\,G_k\,K\,]^T \right) \odot G \right) \ ,$$

---

[1] Symmetrisation over the node / vertex indices given by $\text{symmetric}(M) = M_{ij} + M_{ji}$ for $i, j = 1, \ldots, P$ for a square matrix $M \in \mathbb{R}^{P \times P}$.

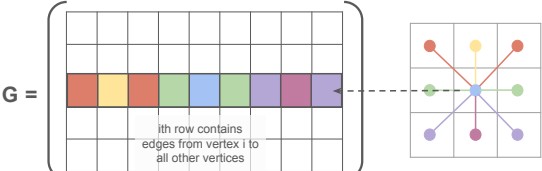 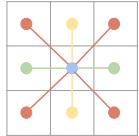 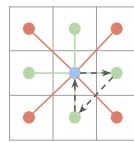

**(a)** Horizontally flip symmetry preserving Graph matrix. Its $i^{th}$ column is given by the horizontally flip symmetric graph centered around the $i^{th}$ patch vertex (here $i = 5$).

**(b)** Horizontal and vertical flip preserving graph. Additional rotation invariance requires green = yellow.

**(c)** Rotation preserving. Flip symmetry broken by directed triangle sub-graphs.

**Figure 2:** Composition choices of the graph matrix $G \in \mathbb{R}^{P \times P}$ for $P = 9$ to preserve different symmetries. Same colours in $G$ represent shared weights. In (c) flips change the orientation of directed triangles i.e. clockwise to anti-clockwise while $90°$-rotations preserve the latter.

where $\odot$ is the point-wise Hadamard product. Here, the **attention score matrix $\Gamma$ is interpreted as the learned effective graph representation.** Analogous to equation 2, the grid symmetries are imposed by a graph topology matrix $G$ which breaks permutation invariance of the standard self-attention (equation 1). Assuming that the image is split into $P$ patches, the graph matrices $G_{k,v,q} \in \mathbb{R}^{P \times P \times d_f}$ and $G \in \mathbb{R}^{P \times P \times \# \text{ heads}}$ are to be chosen for each feature/head from either of the different symmetry preserving graph matrices depicted in Figure 2. The matrix and point-wise multiplication in equation 3 is applied per each feature and head dimension, respectively. Notably, GSA reduces to the conventional attention mechanism when the underlying graph topology matrix G only contains self-loops, i.e. $G$ being the identity matrix.[2]

In Figure 2, we highlight variants of a 2D grid topology matrix $G$ preserving different symmetries, where *same* colors represent a *shared* weight. For example, when using horizontal in 2a, vertices and edges of the same color are transformed into each other, creating a symmetry. Other transformations do not produce this effect. Now, we define G formally. For more technical details, see Appendix A. Assume that the distances are measured w.r.t. a specific vertex, e.g. the center one in Figure 2, and edges can be viewed as vectors. Then, pick $G \in \mathbb{R}^{P \times P}$ such that a shared weight is present in $G$:

- 2(a). When horizontal component of edges have the same magnitude (Horizontal flip-preserving)
- 2(b). When the magnitude of the edges is same (Horizontal and vertical flip-preserving)
- 2(c). When the distance between vertices is consistent. (Rotation preserving)

**Flip symmetry breaking layer which preserves rotation symmetry.** To preserve meaning of the direction of the agent, it is imperative to break flip-symmetries, as such symmetries interchange the sense of left / right or up / down. To do so, we consider directed triangle sub-graphs. In Figure 2(c), flips and rotation acting are symmetries as they map the graph $G$ to itself. However, the directed triangle changes orientations from clock-wise to counter-clockwise for flips, while it remains the same for rotations. When applying this insight upon the learned graph representation $\Gamma$ we can construct a flip symmetry breaking layer that still preserves the rotation symmetry. In particular, we sum over distinct directed triangle sub-graphs of the attention score matrix $\Gamma$:[3]

$$\Gamma^{\text{rot}}(Q, K)_{ij} = \Theta^{(i \to j \to k)} \Gamma(Q, K)_{ij} + \Theta^{(j \to k \to i)} \Gamma(Q, K)_{jk} + \Theta^{(k \to i \to j)} \Gamma(Q, K)_{ki} , \quad (4)$$

where $\Theta^{(j \to k \to i)}$ are weights chosen in a particular way depending on the angels between edges between vertices $j, k, i$. The resulting new graph score matrix $\Gamma^{\text{rot}}$ distinguishes between flips of the input data while remaining invariant upon $90°$ rotations.

**Proposition 3.1 (Symmetry Guarantee)** *The GSA mechanism (equation 3) represents a symmetry-preserving module. It may be both invariant and/or equivariant w.r.t. symmetries of the input. The corresponding symmetry is dictated by the various graph selections. To achieve rotation invariance, the subsequent application of equation 4 is necessary. For **invariance** the token embedding i.e. the artificial $(P\text{-}1)^{th}$ patch is utilized at the output. Due to this mechanism, self-attention (equation 1) is permutation invariant. **Equivariance** is achieved for the P-dimensional patch information of the output, i.e. not related to the token embedding.*

---

[2]In the appendix, we propose other variants of the GSA, e.g., with anti-symmetrization in equation 6.

[3]As a preliminary step, this requires graph matrices $G$, as depicted in Figure 2(c).

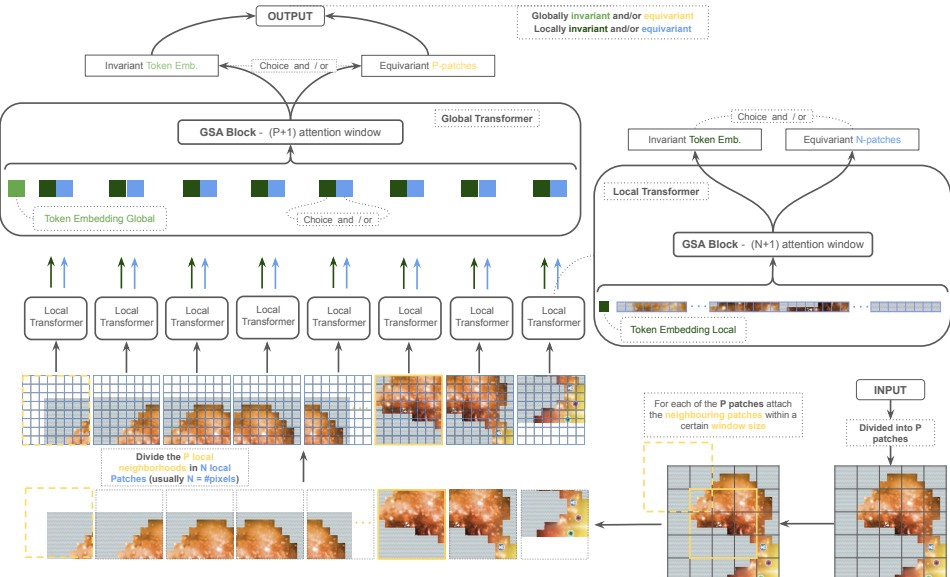

**Figure 3:** SiT model architecture wit local and global GSA modules.

**Explicit & Adaptive Symmetry Breaking**. The graph matrices $G$ and $G_{k,v,q}$ at weight initialisation explicitly break the symmetry of self-attention from permutation invariance (PI) or equivariance (PE) to the respective choice, see Figure 2. However, PI or PE may be approximately recovered by GSA during training in a self-supervised manner, as it corresponds simply to the identity matrix $G$, $G_{k,v,q}, = \mathbf{1}$. For example, in Figure 2(b), rotation invariance can be learned if the yellow weights approximately equal the green ones. We choose $G_{k,v,q}$ with $G_v = 1$ for the main experiments.

As far as we know, this symmetry-preserving GSA has not appeared previously. Prior work (Fuchs et al., 2020; Romero & Cordonnier, 2021) discuss only modification of the positional embedding. The latter, is entirely omitted by our approach. Symmetries in (Fuchs et al., 2020; Romero & Cordonnier, 2021) are imposed by addition of the positional embedding to the input, however each subsequent attention layer is still PI invariant w.r.t. to its respective inputs. In contrast, in our approach each GSA layer is individually able to reduce PI invariance to graph symmetries and is thus able to infer spatial 2D information of the input and latent features.

## 4 SYMMETRY-INVARIANT AND EQUIVARIANT TRANSFORMERS

Symmetry invariant transformer (**SiT**) is a vision transformer that employs the GSA mechanism (equation 3), and optionally equation 4, both locally as well as globally. See Figure 3 for a visualization. We refer to attention applied to entire image patches as "global". On the other hand, "local" attention is applied to a specific patch or its surrounding neighborhood.

Invariance is obtained by the same mechanism as the permutation-invariance of self-attention, i.e. the token-embedding is added as the $(P+1)^{th}$ patch to the input. Since the token embedding does not change under transformations of the input data, the transformer model remains invariant if only the token embedding is considered at the output. In contrast, the representation along the patch dimensions changes under symmetry transformations of the input; however there is a specific way in which one can trace that property throughout the transformer model; we refer to the latter as the equivariant patch-representation. For a more formal argument, please see appendix E.

Based on the above discussion, an invariant SiT forward propagates only the symmetry-invariant token embedding to subsequent layers. In contrast, equivariant SiT (**SeT**) forward propagates the equivariant patch-representation both locally and globally. Symmetry-invariant-equivariant Transformer (**SieTs**) is both local and global invariant and equivariant. The global symmetry are a result of the local attention and the global attention mechanism. For example, a global 90° rotation is can be thought of as the rotation of the position of the patches (global GSA symmetry) and additionally local rotation of every single patch on pixel level (local GSA symmetry), see Figure 9.

**SiT with Preservation of Directions**. Since all flips alter the interpretation of left/right and up/down, only local and global rotation-invariant SiTs maintain the agent's meaning of direction. Additionally,

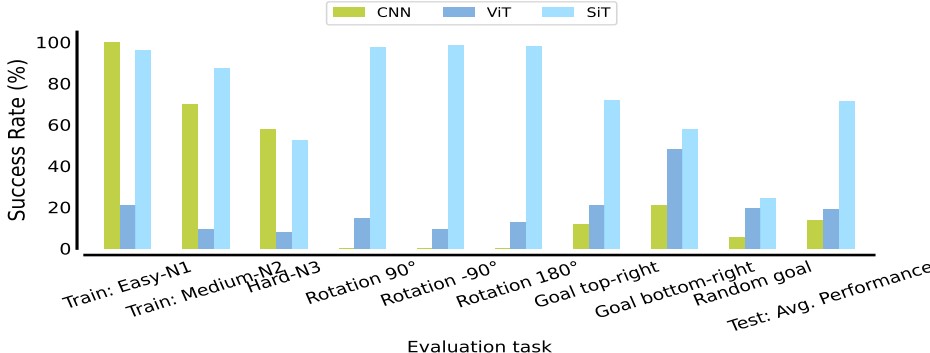

**Figure 4: Comparing SiTs with CNNs and ViTs**, in terms of training and generalization performance on LavaCrossing environments. SiTs substantially outperform both CNNs and ViTs.

Figure 1(c) illustrates that the agent's surroundings shift with local patch rotations. To address this challenge, we enlarge the local attention window across multiple patches. This "softly" breaks local symmetries, hence, only the global rotation persists as an exact symmetry in our empirically tested invariant SiT version in section 5, the one most fitting for RL tasks. Nonetheless, local symmetries may be restored during training in a self-supervised manner.

**Graph Symmetric Dropout**: Moreover, since a conventional dropout function - likely required for large transformer SiT models in vision and language tasks - breaks the inductive bias explicitly. We introduce graph symmetric dropout which preserves the symmetry of the underlying graph. A symmetry preserving dropout for the GSA layer is obtained by setting specific shared weights in the graph matrices $G$, $G_{k,v,q}$ to zero. This statement follows from proposition 3.1.

### 4.1 SCALABILITY OF SITS

ViTs are known to require more working memory (RAM) of the GPU than CNNs, due to the softmax operation (Dao et al., 2022). The local attention mechanism of SiTs is applied to larger effective batch-sizes as the actual batch-size of the input is compounded by the number of total patches of the global attention. Using a larger local attention window only increases this overflow. In our current implementation, SiTs are 2x-5x times slower to execute than ViTs of comparable size. However, this limitation is due to our custom implementation of our neural-net layers (GSA, graph triangulation) and may be resolved by a future custom CUDA implementation as SiTs can outperform ViTs that contain much larger number of trainable parameters than SiTs.

Nonetheless, technical obstacles arise when scaling SiTs to larger image and batch-sizes in image-based RL environments such as Procgen. We address these by modifying the SiT implementation. First, we establish a connection between graph matrices and depth-wise convolutions with graph-weights as kernels. Secondly, to accommodate for an extended local attention window the graph matrix connects pixels over lager distances while the actual attention-mechanism is focused on a smaller patch.

## 5 EMPIRICAL EVALUATION

### 5.1 GRIDWORLD

=

**Environment Details**. The LavaCrossing environment is a standard component of MiniGrid, a Minimalistic Gridworld toolkit (Chevalier-Boisvert et al., 2019). The primary objective of the agent in this environment is to reach the goal position (green square) without falling into the lava river (orange squares). The game is procedurally generated with three levels of difficulty for each map size. Therefore, this environment is suitable for evaluating the combinatorial and out-of-distribution generalization of learned policies in RL. Moreover, we test the generalisation to rotated observations as well as goal changes, see (Figure 5). While, MiniGrid environments are partially observable by default we configure our instances to be fully observable; as well as change the default observation size from $9 \times 9$ to $14 \times 14$ pixels. For further environment details see the appendix F.

| Procgen Task | CNN | E2CNN | E2CNN$'$ | ViT | SiT | SiT$^*$ | SeT | SieT | CNN-UCB$^*$ (9×) |
|---|---|---|---|---|---|---|---|---|---|
| CaveFlyer | 4.0% | 13.4% | 17.7% | -1.8% | **59.7%** | **55.5%** | 4.6% | **34.5%** | 18.0% |
| StarPilot | 36.3% | 28.1% | 29.4% | 6.7% | 31.3% | 31.0% | 38.4% | **42.2%** | 44.6% |
| Fruitbot | 70.8% | 64.0% | 66.1% | 9.7% | 69.8% | 70.5% | 68.9% | **76.0%** | 85.8% |
| Chaser | 10.6% | 13.6% | 15.6% | 9.0% | 35.6% | 45.6% | 50.1% | **54.0%** | 44.6% |
| Average | 30.4% | 29.8 % | 32.2 % | 5.9% % | 49.1 % | **50.6** % | 40.5 % | **51.7** % | 48.2% |

**Table 1: CNN/ViT vs. SiTs on Procgen environments: Caveflyer, Starpilot, Chaser, Fruitbot**. We train with PPO (DrAC) + Crop augmentation for SiT (Sit$^*$, Set, Siet) and compare to the CNN, and E2CNN with Dihedral symmetry group (Wang et al., 2022a) (ResNet with model size $79.4k$ comparable to the SiT - $65.7k$; ViT with 4-layers - $216k$, E2CNN with 4-layers $70.7k$, and E2CNN$'$ with 4-layers $139.2k$ (increased features) ). We do not alter the ResNet architecture of Raileanu et al. (2020) but chose the same hidden-size of 64 as for the SiT as well reduce the number of channels to [4,8,16]. We train over 25M steps. Following Agarwal et al. (2021b), we report the min-max normalized score that shows how far we are from maximum achievable performance on each environment. All scores are computed by averaging over both the 4 seeds and over the 23M-25M test-steps. The UCB-DrAC results with Impala-CNN(×4) ResNet with 620k parameters are taken from Raileanu et al. (2020).

**Evaluation & Results**. The experiments with deep Q-learning (IMPALA (Espeholt et al., 2018)) here focuses on GSA with graph matrices $G_{k,v,q}$. The number of lava rivers generated in the environment is proportional to the difficulty level. We evaluate the out-of-distribution generalization by training the agent on difficulty level 1 and 2 and testing it on levels 1 to 3 , and varying goals unseen during training (Figure 5). As shown in Figure 4, SiTs generalise better than the CNNs even on tasks which do not include

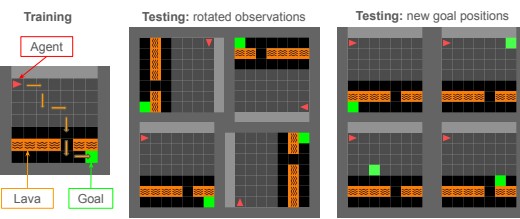

**Figure 5:** Train vs. test observations of the Mini-grid Lavacrossing (easy-N1) environment. We test generalisation of agents to varying goal and starting positions.

rotations - which are reflected inductive by the choice of the attention mechanisms. For an ablation of SiTs, see the appendix G.2.

## 5.2 SCALING SiTs: PROCGEN, ATARI 100K & DM-CONTROL

We demonstrate scalability of SiTs in the widely-studied Procgen benchmark (Cobbe et al., 2020), Atari 100k Bellemare et al. (2013); Kaiser et al. (2020) and DM-control Tassa et al. (2018). For details of the latter see the appendix F.3 and F.2, respectively. The Procgen benchmark corresponds to a distribution of partially observable MDPs (POMDPs) $q(m)$, and each level of a game corresponds to a POMDP sampled from that game's distribution $m \sim q$. The POMDP $m$ is determined by the random seed used to generate the corresponding level. Following the setup from Cobbe et al. (2020), agents are trained on a fixed set of n = 200 levels (generated using seeds from 1 to 200) and tested on the full distribution of levels (generated by sampling seeds uniformly at random from all computer integers). We evaluate test performance on 20 different levels.

Table 1 shows the results for SiT, the equivariant SeT as well as a both invariant & equivariant SieT (all 2 local & 2 global GSA layers) trained with PPO (DrAC (Raileanu et al., 2020)) with crop-data augmentation. Sit$^*$ uses two consecutive sums over triangles of the attention score matrix equation 4.

Invariant SiTs perform well in environments not reflecting the symmetries of the model, e.g. Starpilot, Fruitbot, Chaser are not rotation invariant. We conclude that equivariance is not necessary. However, the combination of invariance and equivariance of SieTs is superior. In SieTs, the invariant and equivariant output of the local and global GSA are concatenated, respectively.

**Results:** SiT almost doubles the performance on the rotation invariant Caveflyer environment w.r.t. to the ResNet (620k weights) of UCB-DrAC (Raileanu et al., 2020). As UCB-Drac uses the rotational data-augmentation for training, we can conclude that SiTs outperform rotational data-augmentation. Overall, our tested SiTs, SeT and SieT models ($\approx$ 70k weights ) substantially outperform the CNN and E2CNN (4-layers) Weiler & Cesa (2021) baselines with similar number of weights while perform comparably to the UCB-ResNet with $9\times$ parameters (620k weights). Notably, all the SiT variants obtain $7 - 9\times$ improvements in performance compared to ViTs.

**Atari 100k:** We evaluate our SieT model on 4 common Atari games, SpaceInvaders, Pong, Breakout, KungFuMaster and find comparable sample-efficiency to the baseline CNN Hessel et al. (2017), see

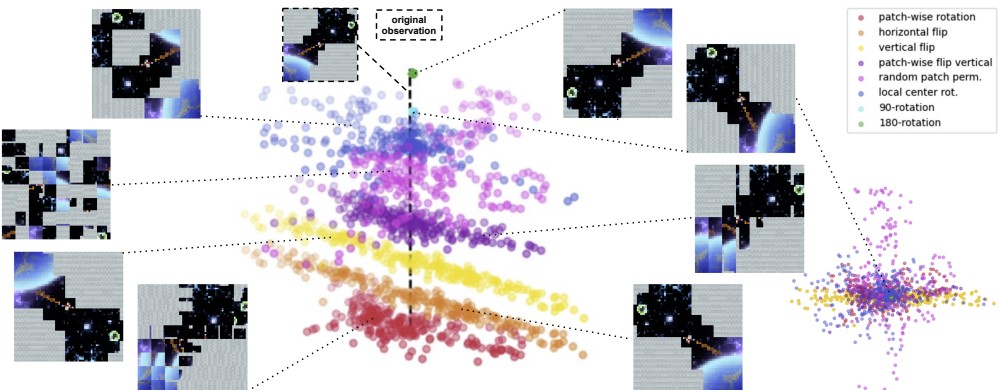

**Figure 6:** PCA of latent space representation of the trained SiT model after 25M steps on Procgen CaveFlyer. We display the PCA of the difference of the latent representation of augmented observations and original ones. 3D-view (center) - vertical separation for illustration purposes - 2D-view i.e. from above (bottom-right); .

Table 3. . DM-control: We employ SieT on top of SACHaarnoja et al. (2018) on the Walker-walk task . Without any hyper-parameter and backbone changes compared to our Procgen setup, SieT has comparable performance to the ViT baseline with > 1M weights (Hansen & Wang, 2021).

**Hyperparameter Sensitivity**. The same limited h-parameter search was performed for SiTs and ViTs. Compared to the ResNet baseline (Raileanu et al., 2020), we employ larger batch-size 96 (instead 8) and PPO-epoch of 2 (instead 3). SiTs don't require tuning except for the batch-size, e.g., a PPO-epoch of 3 works well too. No tuning at all on DM-control and Atari 100k. ViTs generally exhibit suboptimal performance in RL, with the notable exception of Hansen et al. (2021); Tang & Ha (2021). We attribute this that ViTs are less sample-efficient and require different hyperparameter compared to CNNs. As Vits are compute extensive hyperparameter search is practically infeasible. SiTs are applicable to RL tasks as they alleviate both of these caveats.

**Latent representation analysis:** In Figure 6, we present a principal component analysis (PCA) of the latent representation of the policy SiT model. While local symmetries and global flips have been **dynamically broken** during training exact symmetries of **global rotation is preserved**, i.e. all the data-points collapse into (nearly) identical points for the latter. As by design of SiTs, the local attention patch symmetry - *local center rot.* in Figure 6 - is broken "softly" as it is relatively close to the original latent representation. Note that at weight initialisation of SiT, all of the symmetries except permutation invariance are almost exactly preserved.

## 5.3    SiTs beyond RL: Vision Task Ablation

We perform an ablation study of SiTs, SieTs compared to ViTs on a supervised vision task on the CIFAR-10 dataset (Krizhevsky & Hinton, 2009) see Figure 2b.; firstly we compare $G$ in SiT to a conventional position embedding in ViT; secondly we use our SieT model with $G_{k,q}$, $G_v = 1$ to show improved sample efficiency & performance compared to ViTs (Dosovitskiy et al., 2020). SiT and SieT use the horizontal flip symmetry preserving graph matrices.

**Results**. The permutation invariance in original ViT architecture is broken by the use of positional embeddings; in SiT by the graph matrix $G$ in Equation (3). While removing positional embedding to obtain local and global permutation invariance substantially degrades performance  (56% test accuracy), using our graph symmetric attention (80% test accuracy) is superior to using positional embeddings globally (76% test accuracy). Furthermore, Figure 7b shows that SiTs and SieTs reach same performance as ViTs but using $2 - 5\times$ less training epochs.

## 6    Related Work

Symmetry is a prevalent implicit approach in deep learning for designing neural networks with established equivariances and invariances. The literature on symmetries in Vision Transformers (ViTs) Fuchs et al. (2020); Romero & Cordonnier (2021) is relatively limited compared to CNNs (Zhang & Sejnowski, 1988; LeCun et al., 1989; Zhang, 1990), recurrent neural networks (Rumelhart et al., 1986; Hochreiter & Schmidhuber, 1997), graph neural networks (Maron et al., 2019; Satorras et al.,

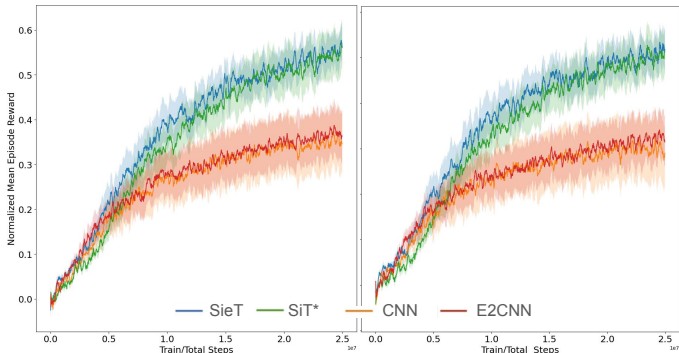

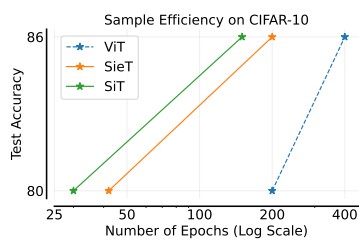

**(b)** SiTs and SieTs are much more sample efficient than ViTs (Dosovit-skiy et al., 2020). The SieT model has 4 local GSA layers and 8 global ones.

**(a)** Train (left) and test curves (right), normlized reward averaged over the StarPilot, SavelFlyer, Chaser, Fruitbot environments.

**Figure 7:** SiTs are comparable to CNNs in terms of sample efficiency on Procgen (a) and outperform Vits (b).

2021a), and capsule networks (Sabour et al., 2017). PI in attention mechanisms and ViTs has been examined in (Lee et al., 2019) and (Tang & Ha, 2021). In contrast, the SiT variants admit different adaptive symmetries other than PI.

The Region ViT method (Chen et al., 2022) divides the feature map into local areas, where each region has tokens that attend to their local counterparts. We use global tokens and use local attention in a neighbouring subset. The method in (Wang et al., 2021) combines local and global attention to reduce complexity, focusing globally on specific windows. For us "global" means standard attention, while "local" pertains to attention within a window.

Conventionally sample efficiency is enhanced by data augmentation (Krizhevsky et al., 2012). Simple image augmentations, such as random crop (Laskin et al., 2020) or shift (Yarats et al., 2021a), can improve RL generalisation performance; in particular when combined with contrastive learning (Agarwal et al., 2021a). SiTs are complementary to data-augmentation.

Algebraic symmetries in Markov Decision Processes (MDP) were initially discussed in (Balaraman & Andrew, 2004) and recently contextualized within RL in (van der Pol et al., 2020). Symmetry-based representation learning (Higgins et al., 2018) refers to the study of symmetries of the environment manifested in the latent representation and was extended to environmental interactions in Caselles-Dupré et al. (2019). These concepts were recently extended in Rezaei-Shoshtari et al. (2022); Mondal et al. (2022). In Weissenbacher et al. (2022), symmetries of the dynamics are inferred in a self-supervised manner; Cheng et al. (2023) discusses time-reversal symmetry. These approaches are mostly complimentary to employing SiTs with equivariance/invariance which may aid the former.

Numerous prior works have demonstrated the exceptional sample efficiency of RL achieved through equivariant methods with CNNs (van der Pol & Welling, 2019; Wang & Walters, 2022). Steerable Equivariant CNNs named E2CNNs Cohen & Welling (2016); Weiler & Cesa (2021) have been widely applied to RL Mondal et al. (2020); Wang et al. (2022a). In contrast, SiTs belong to ViT paradigm, i.e. a distinct new approach to achieve invariance as well as equivariance both locally and globally. Approximately equivariant networks (Wang et al., 2022b) offer a flexible and adaptive approach by imposing constraints on the weights via a regularizer. Rotation invariance steerable convolution in toy-examples is discussed in Zhao et al. (2023). SiTs start from a manifest PI and describe an implicit adaptive mechanism of breaking it and scale to relevant tasks in RL.

## 7 CONCLUSIONS

In this work, we introduced the Graph Symmetric Attention (GSA) mechanism, a symmetry-preserving attention layer that adapts the self-attention mechanism to maintain graph symmetries. We combine GSA with ViTs to propose the novel SiT architecture. By leveraging the interplay of local and global information, SiT achieves inherent out-of-distribution generalization on RL environments. Transformers have significantly advanced natural-language-processing and vision tasks, particularly in scalability. Our work may pave the way for applying these benefits to image-based RL. Additionally, transformers facilitate integration with Large Language Models (LLMs) for multimodal architectures, highlighting their potential in future vision and language-based RL research.

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

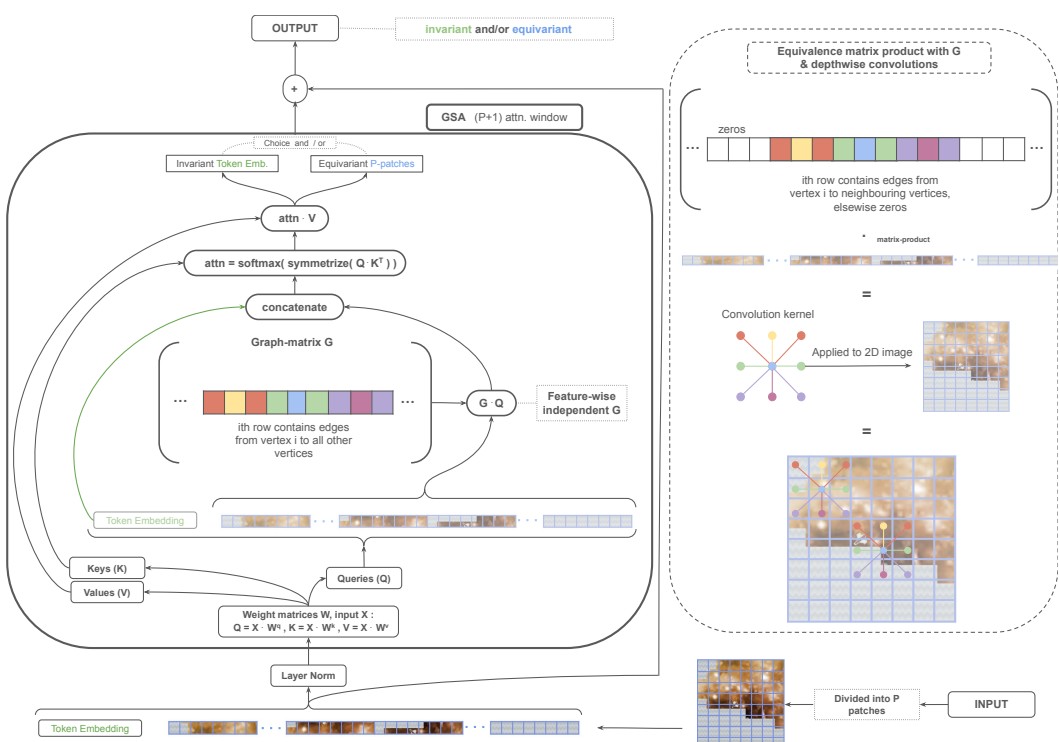

**Figure 8:** GSA module architecture (left), and equivalence of graph matrix and depth-wise convolutions (right). The GSA layer includes Layer-Norm and additive skip connection in addition to the GSA module. We display the GSA variant $G_{k,v,q}$ with $G_{k,v} = 1$.

# A OVERVIEW OF DEFINITIONS: GRAPH SYMMETRIC ATTENTION MECHANISM

We propose the following **G**raph **S**ymmetric **A**ttention (GSA) mechanism

$$\text{GSA}(K, V, Q) = \text{softmax}\left( \frac{1}{\sqrt{d_q}} \Gamma(Q, K) \right) G_v \ V \tag{5}$$

$$\text{with } \Gamma(Q, K) = \text{symmetric}\left( \sigma\left( G_{qk} \ \left( G_q \ Q \ \ [\ G_k \ K\ ]^T \right) + G_b \right) \odot G \right),$$

where $G$, $G_{k,v,q}$, $G_{kq,b} \in \mathbb{R}^{P \times P}$ being the graph matrices described in figure 2 and $\sigma$ is an activation function . The color coding refers to different conceptual implementations, which may be used in combination. When using token embeddings $K$, $V$, $Q$, $\in \mathbb{R}^{P+1 \times d_f}$ thus $G_{k,v,q} \in \mathbb{R}^{P+1 \times P+1 \times d_f}$ and $G$, $G_{kq,b} \in \mathbb{R}^{P+1 \times P+1 \times \#heads}$. In particular, that implies that we apply a different set of graph weights to the feature and head dimensions. See Figure (8) for a visualisation.

Moreover, we propose the a second variant of the GSA$^{a-sym}$ of our attention mechanism which replaces the attention matrix as

$$\text{softmax}\left( \frac{1}{\sqrt{d_q}} \text{sym}\left( \Gamma(Q, K) \right) \right) + \text{softmax}\left( \frac{1}{\sqrt{d_q}} \text{a-sym}\left( \Gamma(Q, K) \right) \right) \tag{6}$$

where we symmetrise and anti-symmetrises over the patch indices of $\Gamma(Q, K)$ respectively, the latter is as in equation 3; and *a-sym* refers to replacing the symmetrisation in equation 14 by anti-symmetrisation. Anti-symmetrisation of a matrix M refers to $M_{ij} \rightarrow M_{ij} - M_{ji}$.

**Empirical Evaluation Omission Overview:**

- On our grid-world environment experiments we found qualitatively that the variants $G_{kq,b}$, $G$ required more h-parameter tuning to show comparable performance to the CNN baseline. We thus removed a quantitative analysis from the paper.[4]

- Adding the anti-symmetrisation equation 6 to the architecture increased generalisation performance on the grid-world environment. However, it requires two softmax operation, which renders it hard to scale; we thus removed a quantitative analysis from the paper.

**More formally of can define e.g. G in Figure 2 (c).** Pick $G \in \mathbb{R}^{P \times P}$ such that it admits a shared weight if the distance between vertices is the same. For more technical details see the appendix A More formally, $G_{ij} = \theta^{(\kappa)}$, $i, j = 1, .., P$ with weights $\theta$ with labels $\kappa = 1, .., \#(\text{unique edge lengths of 2D grid graph})$. The element $G_{ij}$ corresponds to the edge between the $i^{th}$ and $j^{th}$ vertex, i.e. the assigned weight index $\kappa$ is identical if and only if the distance between the $i^{th}$ and $j^{th}$ vertex is the same.

**Rotational Symmetry.** To ensure that the layer solely possesses rotational symmetry, it is essential to disrupt the flip symmetry. This can be accomplished by selecting flip and rotational graph matrices, as depicted in fig. 2 (c), and summing over distinct directed subgraphs with three vertices, i.e., triangles, while assigning weights to each contribution. equation 3 as follows:

$$\text{GSA}_{\text{rot}}(K, V, Q) \quad = \quad \text{softmax}\Big( \frac{1}{\sqrt{d_q}} \sum_{\text{triangle edge} = 1}^{3\,P^2} \Theta_{\text{tri. edge}}\, \Gamma(Q, K)_{\text{tri. edge}} \Big)\, V \qquad (7)$$

$$= \quad \text{softmax}\Big( \frac{1}{\sqrt{d_q}} \Gamma^{\text{rot}}(Q, K) \Big)\, V \quad ,$$

where

$$\Gamma^{\text{rot}}(Q, K)_{ij} \quad = \quad \Theta^{(i \to j \to k)}\, \Gamma(Q, K)_{ij} \; + \; \Theta^{(j \to k \to i)}\, \Gamma(Q, K)_{jk} \; + \; \Theta^{(k \to i \to j)}\, \Gamma(Q, K)_{ki} \; , \; (8)$$

In essence, this implies that for any component of $\Gamma(Q, K)_{\text{edge}}$, two additional entries are added, all weighted with $\Theta's$. The $\Theta's$ are trainable parameters, which will be shared if the angle between two edges of the triangle is identical.

In equation 4, the third vertex of the triangle is chosen as a function of $i, j \mapsto k = T(i, j)$ for a unique map $T$; the weights $\Theta's$ are shared if the angle between edges of the triangle in the square grid is identical. The label $(i \to j \to k)$ denotes the angle at the $j^{th}$ vertex i.e. between the $(ij)$ and $(jk)$ edge. In essence, this implies that for any of the $P^2$-components of $\Gamma(Q, K)$, two additional entries are added, all weighted with $\Theta's$.

A unique triangulation of the square grid is be chosen as follows. Disregarding the heads-dimension for the time being, any entry of the matrix $\Gamma(Q, K)$ can be construed as a connection between a specific patch and another, thereby enabling the drawing of an edge from the former to the latter. This forms the initial directed edge of the triangle, linking two vertices. From the end of the latter, we opt to turn right and proceed to the nearest vertex in the grid, constituting the third vertex of the triangle. The direction of the edges is determined by traversing the triangle. Notably, as flips modify left/right and/or up/down, the aforementioned sum is not invariant under flips; however, it preserves left/right rotations, which transform the grid into itself.

**Duality to convolutional kernels.** See Figure (8) and (2) for a visualisation. The shared nature of the graph-matrix elements make them identical to a 2D-convolutional layer if the $kernel\ size = 2 * image\ size + 1$, for a square image. If a specific kernel size is chosen it corresponds to the graph matrix with zero entries for vertices of larger distance than the kernel size, see Figure (8).

### A.1 ONE-DIMENSIONAL DATA / SEQUENTIAL DATA

While our discussion centers on the 2D grid, GSA may be adapted to 1D data where it ensures shift-symmetry and an optional flip-symmetry.

---

[4]Let us stress that both $G$ and $G_{v,k,q}$ are indeed sufficient to break symmetries, respectively, i.e. to achieve the desired equivariance.

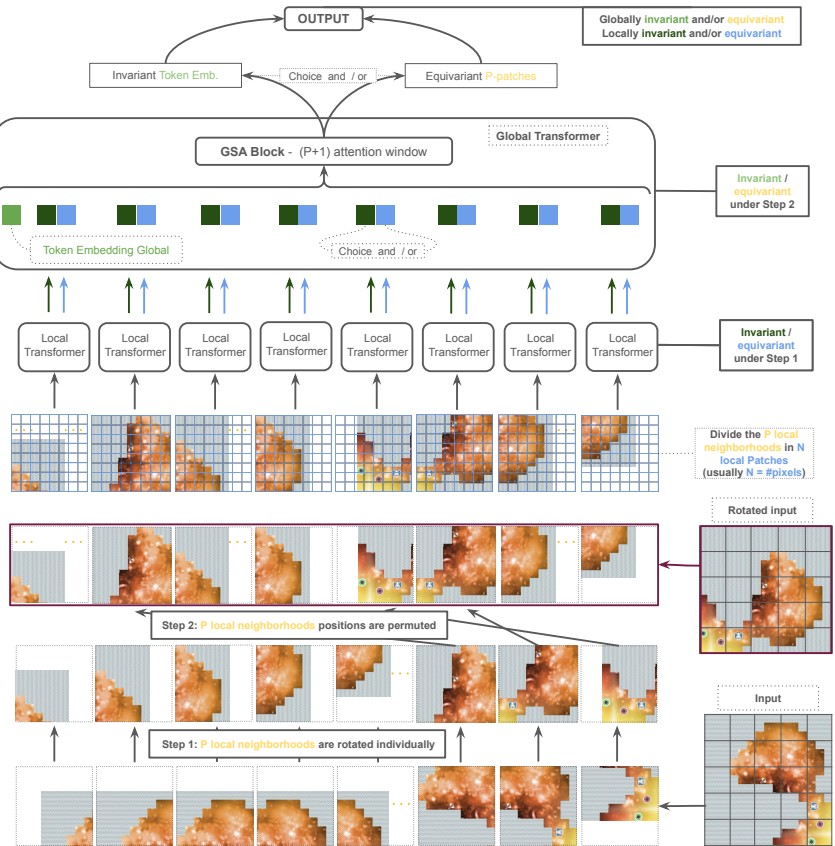

**Figure 9:** Illustration on exact global symmetries of SiTs; obtained as a result of the combined effect of local and global GSA modules.

One may define G formally for the 1D case as. The data points in the 1D data are consider vertices, then assume that the distances are measured w.r.t. a specific vertex, and edges can be viewed as vectors. Then, pick $G \in \mathbb{R}^{P \times P}$ such that a shared weight is present in $G$:

1. When horizontal component of edges have the same magnitude ( flip-preserving)

2. When the magnitude of the edges is same, but direction left/right i.e. the sign is accounted for ( shift symmetric )

Points (1) and (2) above are completely analog to the more complicated 2D case which is proofed in section E. Flip-preserving 1D GSA (1) for sequential data implements time-reversal symmetry, recently discussed in the context of RL Cheng et al. (2023).

**Application of Transformers in RL on Sequence data:** The rise of Transformers in model-based RL Chen et al. (2021); Micheli et al. (2023) opens up another direction which may adapt our approach.

## B  OVERVIEW: SYMMETRY-INVARIANT TRANSFORMER

See Figure (3) for a schematic overview of the SiT architecture and for details on the symmetry of SiTs Figure 9. The local attention window's are processed by a GSA, which then passes only the token embedding dimension onto the global GSA layers. Since the token embedding dimension is invariant under the respective symmetries the global GSA receives a symmetry-invariant input of the local patches.

## C ALGORITHM DETAILS

```python
class GraphSymmetricAttention(Module):
    def __init__(self, dim, num_patches, num_heads):
        self.num_heads = num_heads
        self.qkv = Linear(dim, dim * 3) # Fully-connected layer
        # Generate shareable graph weights
        self.G_weights, self.G_idxs = graph_function(num_patches, dim *
    3)

    def  qkv_SymBreak(self,x):
        # Select indices of weights along dimension
        G = index_select(self.G_weights , 1, self.G_idxs)
        # Matrix mulitply with G; omit token embedding dimension
        y = G@x[:,1:,:]
        return cat([x[:,0:1,:], y],dim=1)

    def forward(self, x):
        B, N, C = x.shape
        qkv = self.qkv_SymBreak(self.qkv(x))
        qkv = qkv.reshape(B, N, 3, self.num_heads, C // self.num_heads).
    permute(2, 0, 3, 1, 4)
        q, k, v = qkv[0], qkv[1], qkv[2]
        attn = (q @ k.transpose(-2, -1))
        # symmetric attenion (if used)
        attn = attn + attn.transpose(-2, -1)
        attn = attn.softmax(dim=-1)
        return (attn @ v ).transpose(1, 2).reshape(B, N, C)
```

**Listing 1:** Pseudocode for GSA (PyTorch-like). Changes relative to self-attention in brown.

```python
def graph_function(num_patches, in_features):
    #compute simple ditance of vertices in square grid
    for k in range(0,num_patches):
        for l in range(0,num_patches):
            for i in range(0,num_patches):
                for j in range(0,num_patches):
                    I = k*num_patches+l
                    J = i*num_patches+j
                    dist[I,J] =  sqrt((k-i)**2 + (l-j)**2)

    #identify equal ditances and associate unique index
    unique = dist.flatten().unique()
    dim = unique.shape[0]
    for i in range(dim):
        mask = (dist == unique[i])
        idxs[mask] = i
    #initialise independent weights
    weights = Parameter(Tensor(in_features,dim))
    return weights , idxs
```

**Listing 2:** Pseudocode for shared weight graph indices (PyTorch-like).

The implementation of the rotation symmetry breaking is a bit lengthy however straightforward. We omit details of how to compute the angles of triangles of vertices on the square grid, and just assert the function "rot_grid_to_idx" here.

```python
class GraphSymmetricAttention(Module):
    def __init__(self, dim, num_patches, num_heads):
        self.rot_SymBreak = Rotation_Symmetry( num_patches, num_heads)

    # ... we only highlight the differnces in the forward pass

    def forward(self, x):
        B, N, C = x.shape
```

```python
        qkv = self.qkv_SymBreak(self.qkv(x))
        qkv = qkv.reshape(B, N, 3, self.num_heads, C // self.num_heads).
    permute(2, 0, 3, 1, 4)
        q, k, v = qkv[0], qkv[1], qkv[2]
        attn = (q @ k.transpose(-2, -1))
        # symmetric attenion (if used)
        attn = attn + attn.transpose(-2, -1)
        attn =  self.rot_SymBreak(attn)
        attn = attn.softmax(dim=-1)
        return (attn @ v ).transpose(1, 2).reshape(B, N, C)

class  Rotation_Symmetry(Module):
    def __init__(self, num_patches, num_heads):
        #compute unique angles of triangles in square grid
        idxs, idxs_angles, dim = rot_grid_to_idx(num_patches)
        self.idxs = idxs
        self.idxs_angles = idxs_angles
        #initialise independent weights
        self.angles =  Parameter(Tensor(num_heads,dim))

    def forward(self, x):
        bs, heads, ps, ps= x.size()
        #index only non-token dimensions
        y = x[:,:,1:,1:].flatten(-2)[:,:,self.idxs]
        #preserve rotation/translation invariances, break flip
    invariances
        angles = index_select(self.angles,1,self.idxs_angles.flatten())
        #multply vertex with corresponding weight and sum over triangle
        y = (y*angles).reshape((bs, heads, ps-1, ps-1,3)).sum(-1)
        #attach token dimension
        y = cat([ x[:,:,:,0:1],cat([x[:,:,0:1,1:], y],dim=2)],dim=-1)
        return   y.reshape(x.shape)
```

**Listing 3:** Pseudocode for GSA with rotation symmetry(PyTorch-like). Changes relative to self-attention in brown.

Lastly, in order to achieve scalability of SiTs we need to establish a connecting between graph matrices and depth-wise convolutions with graph-weights as kernels. The latter implementation is significantly more memory efficient. We note here that the below code is equivalent to multiplication of graph matrix given in Lisitng 1.

```python
def graph_function(kernel_size):
        dist_Mat = torch.zeros((kernel_size,kernel_size))

        #only works correcttly if kernel_size is odd
        i0,j0 = (kernel_size -1) //2, (kernel_size -1) //2
        for i in range(0,kernel_size):
            for j in range(0,kernel_size):
                        distance = math.sqrt((i0-i)**2 + (j0-j)**2)
                        dist_Mat[i,j] =  distance

        unique = torch.unique(dist_Mat)
        idxs = torch.zeros(dist.shape)
        dim= unique.shape[0]
        for i in range(dim):
            mask = (dist == unique[i])
            idxs[mask] = i

        #initialise independent weights
        weights = Parameter(Tensor(in_features,dim))
        return weights , idxs

class GraphSymmetricAttentionEfficient(Module):
```

```
...
def  qkv_SymBreak(self,x):
    # Select indices of weights
    #of shape (in\_features,1, kernel\_size, kernel\_size)
    G = index_select(self.G_weights , 1, self.G_idxs)
    # depthwise convlution with weights G; omit token embedding
dimension
    y = conv2d(x, G, padding= (kernel_size-1)//2, groups=in_features)
    return y
...
```

**Listing 4:** Pseudocode for an efficient GSA (PyTorch-like). Changes relative to Listing (1) and (2) are presented.

## D   GSA - GRAPH SYMMETRIC ATTENTION

In the following $K$, $V$, and $Q$ denote the keys, values, and queries respectively. They are derived from the input $X$: $K = XW^k$, $V = XW^v$, $Q = XW^q$, where $W^q$, $W^k$, and $W^v$ are the corresponding weight matrices, i.e. $V_{ia} = \sum_{=1}^{d_f} X_{ix} W^v_{xa}$ The permutation invariant self-attention layer (Lee et al., 2019) is given by

$$Att(K,V,Q) \quad = \quad f(Q,K) \ \ X \ \ W^v \tag{9}$$

$$\text{with} \ \ f(Q,K) \quad = \quad \tfrac{1}{\sqrt{d_q}}\text{softmax}\left( Q \ \ [X \ \ W^k]^T \right) \ \ , $$

which can be rewritten using explicit indices. One finds

$$Att(K,V,Q)_{qa} \quad = \quad \sum_{i=1}^{P} \ \ \text{softmax}\left( \tfrac{1}{\sqrt{d_q}} f(Q,K)_{qi} \right) V_{ia} \tag{10}$$

$$\text{with} \ \ f(Q,K)_{qi} \quad = \quad \sum_{a=1}^{\#heads} \ \ Q_{qa}[K]^T_{ai} \ \ , \tag{11}$$

For the original self-attention mechanism Dosovitskiy et al. (2020) one simply needs to use a non-fixed $Q = X \ \ W^q$ in equation 11 as

$$Att(K,V,Q)_{qa} \quad = \quad \sum_{i=1}^{P} \tfrac{1}{\sqrt{d_q}}\text{softmax}\left( f(Q,K)_{qi} \right) V_{ia} \tag{12}$$

$$\text{with} \ \ f(Q,K)_{qi} \quad = \quad \sum_{a=1}^{\#heads} \ \ Q_{qa} \, [K]^T_{ai} \ \ , \tag{13}$$

We propose the following three variants of **G**raph **S**ymmetric **A**ttention (GSAention) layer - all of which separately preserves the identical symmetries -

$$\text{GSA}(K,V,Q) \quad = \quad \text{softmax}\left( \tfrac{1}{\sqrt{d_q}}\Gamma(Q,K) \right) \ \ G_v \ V \tag{14}$$

$$\text{with} \ \ \Gamma(Q,K) \quad = \quad \text{symmetric}\left( G_q \ Q \ \ [G_k \, K]^T \right) \ \ , $$

and

$$\text{GSA}(K,V,Q) \quad = \quad \text{softmax}\left( \tfrac{1}{\sqrt{d_q}}\Gamma(Q,K) \right) V \tag{15}$$

$$\text{with} \ \ \Gamma(Q,K) \quad = \quad \text{symmetric} \, \sigma\left( G_{qk} \ \ \left( Q \, [K]^T \right) \ + \ G_b \right) \ \ , $$

and moreover

$$\text{GSA}(K,V,Q) \quad = \quad \text{softmax}\left( \tfrac{1}{\sqrt{d_q}}\Gamma(Q,K) \right) \ \ V \ \ W^v \tag{16}$$

$$\text{with} \ \ \Gamma(Q,K) \quad = \quad \text{symmetric}\left( Q \ \ [K]^T \right) \odot G \ \ , $$

In index notation equation 14 becomes

$$GSA(K,V,Q)_{qa} \;=\; \sum_{i=1}^{P} \; \frac{1}{\sqrt{d_q}} \text{softmax}\Big(\Gamma(Q,K)_{qi}\Big) \sum_{j=1}^{P} G_{vij}\, V_{ja} \tag{17}$$

$$\text{with } \bar{\Gamma}(Q,K)_{qi} \;=\; \sum_{a=1}^{\#heads} \sum_{k=1}^{P} G_{qqk}\, Q_{ka} \sum_{j=1}^{P} G_{kij}\,[K]^T_{aj}\;, \tag{18}$$

$$\Gamma(Q,K)_{ij} \;=\; \bar{\Gamma}(Q,K)_{ij} \,+\, \bar{\Gamma}(Q,K)_{ji}\;. \tag{19}$$

Moreover, equation 15 in detailed index notation is given by

$$\bar{\Gamma}(Q,K)_{qi} \;=\; \sigma\Big( \sum_{a=1}^{\#heads} \sum_{j=1}^{P} G_{qkqj}\, Q_{ja}\,[\,K\,]^T_{ai} + G_{bqi}\Big)\;. \tag{20}$$

## E    PROOFS OF INVARIANCE

Permutation matrices $\mathcal{P}$ are orthogonal matrices. An orthogonal matrix is a square matrix whose transpose is equal to its inverse, i.e. $\mathcal{P}^T = \mathcal{P}^{-1}$. A permutation matrix is a square matrix obtained by permuting the rows and columns of an identity matrix. It represents a permutation of the elements in a vector or a rearrangement of the columns and rows of another matrix. Since permuting the rows and columns of an identity matrix results in swapping rows and columns, the transpose of a permutation matrix is equal to its inverse. Therefore, permutation matrices are orthogonal matrices.

The main theoretical claims of this work are summarized in proposition 3.1 , repeated here

**Proposition E.1 (Symmetry Guarantee)** *The GSA mechanism (equation 3) represents a symmetry-preserving module. It may be both invariant and/or equivariant w.r.t. symmetries of the input. The corresponding symmetry is dictated by the various graph selections. To achieve rotation invariance, the subsequent application of equation 4 is necessary. For* **invariance** *the token embedding i.e. the artificial $(P\text{-}1)^{th}$ patch is utilized at the output. Due to this mechanism, self-attention (equation 1) is permutation invariant.* **Equivariance** *is achieved for the P-dimensional patch information of the output, i.e. not related to the token embedding.*

First of all let us emphasise that we have implicitly empirically tested the validity of this claim in the various RL and supervised experiments in this work Let us discuss above claims in several steps:

**The attention mechanism equation 1 is permutation invariant (PI).**    Let $\mathcal{P}$ be a permutation of the input, i.e. $Q, K, V \rightarrow \mathcal{P}Q, \mathcal{P}K, \mathcal{P}V$.

$$Att(\mathcal{P}K, \mathcal{P}V, \mathcal{P}Q)_{qa} \;=\; \sum_{i=1}^{P} \text{softmax}\Big( \frac{1}{\sqrt{d_q}} \Sigma(\mathcal{P}Q, \mathcal{P}K)_{qi} \Big) \sum_{j=1}^{P} \mathcal{P}_{ij} V_{ja} \tag{21}$$

$$\text{with } \Sigma(\mathcal{P}Q, \mathcal{P}K)_{qi} \;=\; \sum_{a=1}^{\#heads} \sum_{l=1}^{P} \mathcal{P}_{ql} Q_{la} \Big[ \sum_{j=1}^{P} \mathcal{P}_{ij} K \Big]^T_{aj}\;, \tag{22}$$

with $[\mathcal{P}\,K]^T = K^T \mathcal{P}^T = K^T \mathcal{P}^{-1}$ na d by using that the permutation matrix can be pulled out of the softmax-function as it is not affected by it one finds

$$\begin{aligned} Att(\mathcal{P}K, \mathcal{P}V, \mathcal{P}Q)_{qa} \;&=\; \sum_{i=1}^{P} \sum_{l=1}^{P} \mathcal{P}_{ql} \Big( \frac{1}{\sqrt{d_q}} \text{softmax}\,\Sigma(Q,K)_{lm} \Big) \sum_{m=1}^{P} \mathcal{P}^{-1}_{mi} \sum_{j=1}^{P} \mathcal{P}_{ij} V_{ja} \\ &=\; \sum_{j=1}^{P} \sum_{l=1}^{P} \mathcal{P}_{ql} \,\text{softmax}\Big( \frac{1}{\sqrt{d_q}} \Sigma(Q,K)_{lj} \Big) V_{ja} \\ &=\; \sum_{l=1}^{P} \mathcal{P}_{ql}\, Att(K,V,Q)_{la} \end{aligned} \tag{23}$$

where we have used that $\sum_{i=1}^{P} \mathcal{P}_{mi}^{-1} \mathcal{P}_{ij} = \delta_{mj}$, where $\delta$ is the Kronecker delta function, i.e. a formal way of writing the identity matrix. **We have showed that the attention mechanism is permutation equivariant. Invariance follows form the observation that when a token embedding is added it is not affected by the permutation matrix which only acts on the patches.** Thus it remains invariant as

$$Att(\mathcal{P}K, \mathcal{P}V, \mathcal{P}Q)_{q=P+1\,a} = Att(K, V, Q)_{q=P+1\,a} \tag{24}$$

**The graph symmetric attention mechanism equation 3 represents a symmetry-preserving , i.e. it can be both invariant and equivariant w.r.t. symmetries of the input. The corresponding symmetry is dictated by the various graph selections.** Let $\mathcal{P}$ be a permutation of the input ,i.e. $Q, K, V \to \mathcal{P}Q, \mathcal{P}K, \mathcal{P}V$ then

$$GSA(\mathcal{P}K, \mathcal{P}V, \mathcal{P}Q)_{q\,a} = \sum_{i=1}^{P} \text{softmax}\Big( \frac{1}{\sqrt{d_q}} \Gamma(\mathcal{P}Q, \mathcal{P}K)_{q\,i} \Big) \sum_{j=1}^{P} G_{vil} \sum_{l=1}^{P} \mathcal{P}_{lj} V_{j\,a} \tag{25}$$

$$\text{with } \Gamma(\mathcal{P}Q, \mathcal{P}K)_{q\,i} = \sum_{a=1}^{\#heads} \sum_{k=1}^{P} G_{qqk} \sum_{l=1}^{P} \mathcal{P}_{km} Q_{m\,a} \sum_{j=1}^{P} G_{kij} \sum_{l=1}^{P} \mathcal{P}_{jl} [K]_{a\,l}^{T} , \tag{26}$$

First of all note that since $\mathcal{P}\,G - G\,\mathcal{P} \neq 0$ , i.e. they do not commute, thus GSA is not permutation equivariant (invariant).

**Definition 1 (Graph Matrix)** *The symmetric graph matrix $G \in \mathbb{R}^{P} \times \mathbb{R}^{P}$ are defined as having a shared weight at entry $G_{ij} = G_{ji}$ if*

- *the distance of the i-vertex to the j-vertex in the square grid of the 2D-image have same length. This leads to a flip and rotation invariant graph matrix.*

- *the horizontal distance of the i-vertex to the j-vertex are the same, and the vertical distance is zero. This leads to a horizontal mirror flip graph.*

Thus a permutation $\mathcal{P}^s$ does commute with the graph matrix if and only if it maps shared weights of G to each other; then $\mathcal{P}^s\,G - G\,\mathcal{P}^s = 0$. Then one can rewrite the above expression as

$$GSA(\mathcal{P}^s K, \mathcal{P}^s V, \mathcal{P}^s Q)_{q\,a} = \sum_{i=1}^{P} \text{softmax}\Big( \frac{1}{\sqrt{d_q}} \Gamma(\mathcal{P}^s Q, \mathcal{P}^s K)_{q\,i} \Big) \sum_{l=1}^{P} \mathcal{P}_{il}^s \sum_{j=1}^{P} G_{vlj} V_{j\,a}$$

$$\text{with } \Gamma(\mathcal{P}^s Q, \mathcal{P}^s K)_{q\,i} = \sum_{a=1}^{\#heads} \sum_{l=1}^{P} \mathcal{P}_{qk}^s \sum_{k=1}^{P} G_{qkm} Q_{m\,a} \sum_{l=1}^{P} \mathcal{P}_{ij}^s \sum_{j=1}^{P} G_{kjl} [K]_{a\,l}^{T} , \tag{27}$$

The remaining steps are analog to the one for the conventional attention. One finds that

$$GSA(\mathcal{P}^s K, \mathcal{P}^s V, \mathcal{P}^s Q)_{i\,a} = \sum_{l=1}^{P} \mathcal{P}_{ij}^s GSA(K, V, Q)_{j\,a} \tag{28}$$

$$GSA(\mathcal{P}^s K, \mathcal{P}^s V, \mathcal{P}^s Q)_{i=P+1\,a} = GSA(K, V, Q)_{i=P+1\,a} \tag{29}$$

**which states the equivariance of GSA and invariance when a token embedding is used at dimension $P + 1$ with respect to the symmetry preserving permutation $\mathcal{P}^s$.**

Two remaining points are twofold. First, the derivation above holds in particular for the symmetrisation (anti-symmetrisation) of $\Gamma$ . Second, to see that graphs matrices with particular choices of shared weights lead to the desired symmetries of the 2D-grid we refer the reader to a visual proof given in figure 2. The proof of equivariance for the setting with $G_{kq}, G_b$ is analog. The case for $G$ one notes that a $P \times P$-matrix $\sum_{i=1}^{P} \mathcal{P}_{ki}^s M_{ij} * G_{ij} = M_{ij} * G_{ij}$ if and only if $\mathcal{P}^s$ only permutes shareable weights of the graph matrix, i.e. thus $\mathcal{P}^s$ needs to obey the symmetry properties. The rest of the steps to conclude the proof are analog as above.

**To achieve rotation invariance, the subsequent application of equation 4 is necessary.** Let $\Gamma_{ij}$ be graph $P \times P$-matrix obeying rotation and flip symmetries of an underlying square grid, and let $\mathcal{T}(i,j) \mapsto k$ be unique mapping for every tuple $(i,j)$ to the vertex $k$; i.e. such that $(i,j,k)$ forms a triangle in the grid. Moreover $\mathcal{T}$ is such that the angles of the resulting triangle only depend on the distance between i-j., and the orientation in clock-wise starting from $i \to j \to k \to i$. The weights $\Theta(ij, jk)$ are shared if the angles in the triangle are equal.

$$\Gamma^{\text{rot}}(Q,K)_{ij} = \Theta^{(i \to j \to k)}\,\Gamma(Q,K)_{ij} + \Theta^{(j \to k \to i)}\,\Gamma(Q,K)_{jk} + \Theta^{(k \to i \to j)}\,\Gamma(Q,K)_{ki} \ , \ (30)$$

The notation is such that $\Theta^{(i \to j \to k)}$ means corresponding to the angle between the edges i-j and j-k. Then we may rewrite the expression from the main text more concisely as where $\mathcal{T}(i,j) = k$. Any transformation which maps equal distance edges to each other will leave $\Gamma(Q,K)_{ij}, \Gamma(Q,K)_{jk}, \Gamma(Q,K)_{ki}$ invariant, respectively. However more specifically flip transformation change the meaning of clock-wise and anti-clockwise and thus

- $\Theta^{(i \to j \to k)} \mapsto \Theta^{(k \to i \to j)}$,

- $\Theta^{(j \to k \to i)} \mapsto \Theta^{(j \to k \to i)}$ ,

- $\Theta^{(k \to i \to j)} \mapsto \Theta^{(i \to j \to k)}$.

Thus equation 30 does not preserve flip transformation. While rotations leave the $\Theta's$ invariant. This concludes the proof of the proposition.

## F  HYPER-PARAMETERS - MAIN EXPERIMENTS

Minigrid - Lavacrossing We employ our invariant SiTs on top of IMPALA Espeholt et al. (2018) implementation based on torchbeast (Küttler et al., 2019). For the experiments involving SiTs and ViTs, we do not employ any hyper-parameter tuning compared to the CNN baseline Jiang et al. (2021) - hyperparameters can be found in that reference. We use the stated number of local and global GSA with an embedding dimension of 64 and 8 heads. The attention window of the global GSA is the entire image, i.e $14 \times 14$ (pixels) which corrsponds to $P = 196$ patches ; while locally we choose patch-size of 5 (pixels), i.e. a attention window of 5x5 (pixels) or $P = 25$ patches. We use rotation invariant GSA both on the local as well as on the global level.

While, MiniGrid environments are partially observable by default we configure our instances to be fully observable; as well as change the default observation size $9 \times 9$ pixel to $14 \times 14$ pixel. The latter is done by rendering the environment observation and then down-scaling to the respective size. Moreover, the default action space only allows the agent to turn left, turn right, or move forward, which requires the agent to keep track of its direction while navigating. To make the environment more accessible , we modify the action space, enabling the agent to move in all four candidate directions i.e. to move left, right, forward and backward. For Minigrid, we selected a 14x14 global image size as this is the minimum resolution at which the direction of the triangular shaped agent can be discerned when downscaling the RGB rendering of the environment. Smaller resolutions fail to capture this detail, while larger ones are feasible but not necessary. The local neighborhood size in Minigrid (in pixels) is required to be odd, so 3x3 and 5x5 are the smallest viable options.

**Architecture Details**. For the CNN baseline, we use two convolutions layers two fully connected ones. The ViT and SiT and architectures both employ a skip connection $x \to x + \text{Att}(x)$ and $x \to x + \text{GSA}(x)$, respectively. As in (Beyer et al., 2022), we modify the Vi architecture of Dosovitskiy et al. (2020) by not using a multi-layer perceptron (MLP) after each attention layer. Our goal is to encounter the most simple functional setting incorporating the attention mechanism. We use one embedding fully-connected layer and use a patch-embedding. We employ our invariant SiTs on top of IMPALA Espeholt et al. (2018) implementation based on torchbeast(Küttler et al., 2019). For the experiments involving SiTs and ViTs, we do not employ any hyper-parameter tuning compared to the CNN baseline Jiang et al. (2021). Qualitatively, even when trying to tune ViTs by running a hyper-parameter sweep, we could not improve their performance by more than a factor of two.

| Suite / Algorithm | Model | Layers | Channels | hidden dim. |
|---|---|---|---|---|
| | *CNN* | 3 ResNet (Raileanu et al. (2020)) | [4,8,16] | 64 |
| | *ViT* | 4 Attn. Layers (Hansen & Wang (2021)) | 64 | 64 |
| | *E2CNN* | 4 E2CNN Layers (Wang et al. (2022a)) | [2,4,8,8] | 64 |
| **Procgen** | *E2CNN'* | 4 E2CNN Layers (Wang et al. (2022a)) | [2,4,8,16] | 64 |
| *Raileanu et al.* (2020) | SiT | 2 local GSA, 2 global GSA | 64 | |
| | SiT* | 2 local GSA, 2 global GSA | 64 | 64 |
| | SeT | 2 local GSA, 2 global GSA | 64 | 64 |
| | SieT | 2 local GSA, 2 global GSA | 64 | 64 |
| | UCB-CNN | 3 ResNet (Raileanu et al. (2020)) | [16,32,32] | 256 |
| **DM−control** Hansen&Wang (2021) | SieT | 2 local GSA, 2 global GSA | 64 | 64 |
| **Atari 100k** | CNN | data-efficient Hessel et al. (2017) | $[32,64]$ ( $\approx 880k$) | 256 |
| Hessel et al. (2017) | SieT | 1 local GSA, 2 global GSA | $128$ ($\approx 309k$) | 256 |

**Table 2:** Architecture and hyper-parameter choices for Procgen and DM-control. UCB-CNN is taken from Raileanu et al. (2020). All Sit variants contain one initial conv. layer with shared graph weights and a subsequent max. pooling layer to reduce the dimensionality of the problem. For the Atari experiments we use the unchanged hyper-parameter setting of Rainbow Hessel et al. (2017).

## F.1 PROCGEN EXPERIMENTS

We train with PPO (DrAC) + Crop augmentation for SiT (Sit*, Set, Siet) and compare to the CNN (ResNet with model size $79.4k$ comparable to the SiT - $65.7k$; ViT with 4-layers - $216k$ ). For parameter details see table (2).

We do not alter the ResNet architecture of Raileanu et al. (2020) but chose the same hidden-size of 64 as for the SiT as well reduce the number of channels to [4,8,16]. We train over 25M steps. Following Agarwal et al. (2021b), we report the min-max normalized score that shows how far we are from maximum achievable performance on each environment. All scores are computed by averaging over both the 4 seeds and over the 23M-25M test-steps. We also report UCB-DrAC results with Impala-CNN($\times 4$) ResNet with 620k parameters, taken from Raileanu et al. (2020).

## F.2 DM CONTROL

We train SieT - Without any hyper-parameter and backbone changes compared to our Procgen setup - wit SAC Haarnoja et al. (2018) for 500k steps. No data-augmentation is used. SieT has comparable performance to the ViT baseline with $> 1$M weights (Hansen & Wang, 2021). For parameter details see table (2).

## F.3 ATARI 100K

The Atari 100k benchmark Kaiser et al. (2020), comprising 26 Atari games Bellemare et al. (2013), spans various mechanics and evaluates a broad spectrum of agent capabilities. In this benchmark, agents are restricted to 100k actions per environment, approximating 2 hours of human gameplay. For context, typical unconstrained Atari agents undergo training for 50 million steps, signifying a 500-fold increase in experience.

Current standards in Atari 100k for search-based methods include MuZero Schrittwieser et al. (2020) and EfficientZero Ye et al. (2021), and recently a transformer based world-model approach Micheli et al. (2023). Image-based SiTs may compliment those methods. In particular the latter may benefit from GSA, see our discussion in section A.1.

Our goal here is not to show a comparison to state-of-the-art on Atari 100k in terms of sample-efficiency but that SiTs may be trained with ease on-top of standard algorithms. Thus, for our Atari 100k experiments we **use the unchanged hyper-parameter setting of Rainbow Hessel et al. (2017)**, both for the baseline model - data-efficient CNN - as well as the algorithm for both CNN and our SieT. The Siet model is the same as in the Procgen experiments but with one less local GSA layer and increased embedding dimension, see Table 2. **The baseline CNN has approximately 3x more weights than SieT.**

| Atari Game | CNN | SieT |
|------------|------|------|
| SpaceInvaders | 344 | 366.3 |
| BreakOut | 4.3 | 4.97 |
| Pong | -19.07 | -20.04 |
| KungFuMaster | 8942.2 | 5696.7 |

**Table 3:** CNN vs. SiTs on Atari 100k benchmark environments: SpaceInvaders, BreakOut, Pong, KungFuMaster. We train with Rainbow Hessel et al. (2017) and compare to the ddtat-effcient CNN ($\approx 880k$ weights ) about $3\times$ larger than our SieT - $\approx 309k$. The number model parameters vary for different environments, however the factor in between CNN baseline and SieT is consistently $\approx 3\times$. We present absolute scores, averaged over evaluation after 80k,90k,100k train-steps and 3 seeds, respectively.

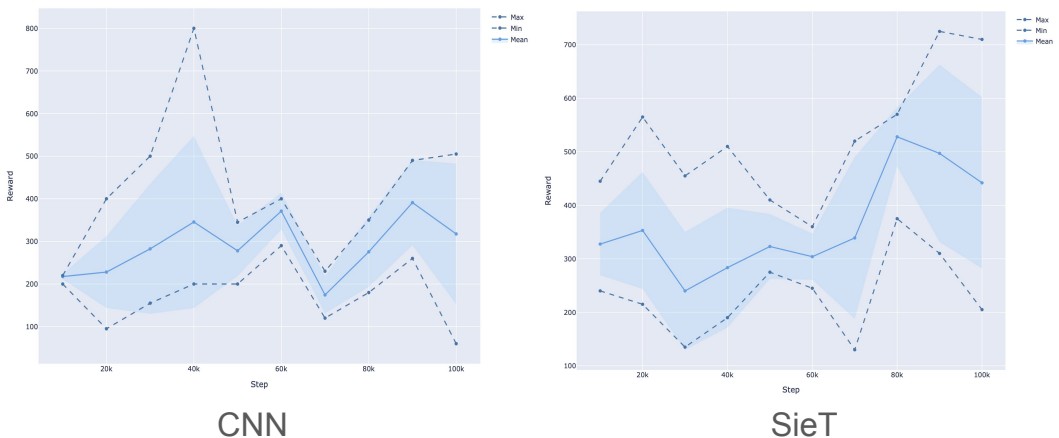

**Figure 10:** Example evaluation curves during training of SpaceInvaders seed 124.

## F.4 TECHNICAL IMPROVEMENTS

Moreover, another minor modification to the architecture lead to a compute speed-up, which is not to use a token embedding in the local attention layer but rather use a depth wise graph-like convolution with kernel-size and stride equal to the patch-size as a last layer. This is common practice for ViTs i.e. by using normal convolutions that way. It is easy to see that our choice also preserves the symmetries of the graph-matrix. So, concludingly using a token embedding is not the only architectural choice which leads to a preservation of symmetries in SiTs.

First, we establish a connection between graph matrices and depth-wise convolutions with graph-weights as kernels. The latter implementation is significantly more memory efficient and faster. Secondly, in order to accommodate for an extend local attention window we do use a graph matrix which connects pixels over lager distances while keeping the actual attention-mechanism focused on a smaller patch.

Thirdly, rather trivially one may scale down the original image size from $64 \times 64$ pixels to $32 \times 32$. This can be done by simple scaling the image, or by using one initial depth-wise convolutional layer with graph-like weights to preserve symmetry plus a subsequent Max-Pooling operation, or by simply using every second pixel of the input-image.

Given the dominance of ViTs in Vision and Transformers in NLP, it's plausible that improvements in Transformer technology will similarly revolutionize vision-based RL, with ViTs becoming predominant. Recent technical advancements, such as efficient Transformers Dao et al. (2022), which offer up to a 10x performance boost may lead the way to a brad adaption of SiTs in RL. As the latter ensures that our sample efficient symmetry-invariant vision transformer becomes rather light-weight.

**CIFAR10 - supervised ablation study:** For the baseline we use a ViT Dosovitskiy et al. (2020) , embedding dimension of 512 , 4 layers, 16 heads, and one local attention layer with the same settings. The SiT has a the graph matrix $G$ added.

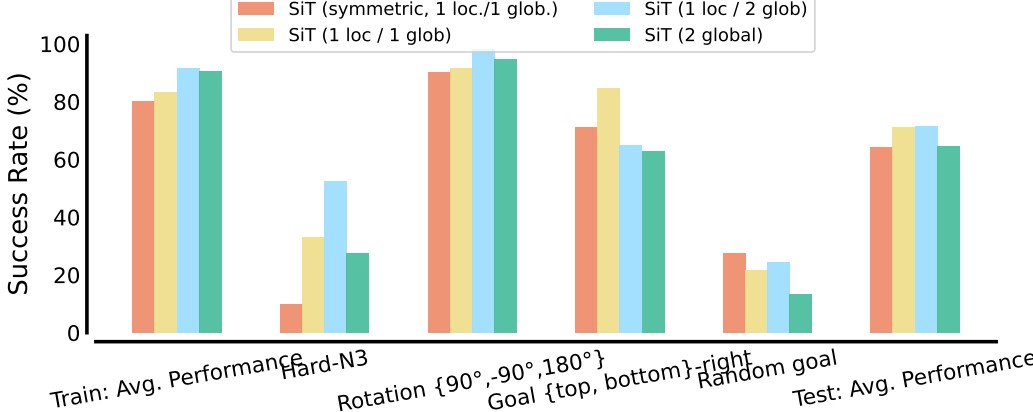

**Figure 11:** Ablations study SiT for different number of local and global layers as well as using the symmetric part of SiT*, defined in the appendix.

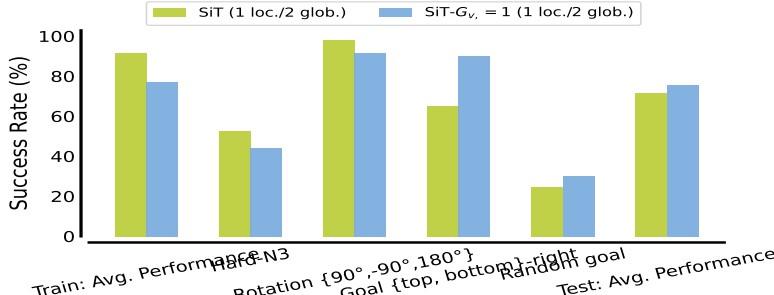

**Figure 12:** Ablations for using the symmetric part of SiT*, defined in the appendix. We ablate by $G_{v,}$ by setting $G_{v,} = 1$ with one local and two global symmetric attention modules equation 6 and rotation invariance.

## G  ADDITIONAL EXPERIMENTS AND ABLATIONS

### G.1  ABLATION: LOCAL AND GLOBAL SYMMETRIES IN ATTENTION FOR IMAGE UNDERSTANDING

In this section we present an ablation study which addresses the impact of local attention fields in SiTs, see results in Figure 11. For the performance comparison of the attention mechanism with both symmetrisation and anti-symmetrisation (Equation 6) in SiTs, we restrict ourselves to provide qualitative results. Performing several experiments with two global layers, of either symmetrisation and anti-symmetrisation or both present, we conclude that indeed the later admits the best relative performance as well as generalisation to the hard task, of all SiT setup with only global layers. However, due to the second softmax in the attention it is more memory consuming and thus applying it in combination with a the local layers is not feasible. Memory efficient attention mechanism Dao et al. (2022) have been developed recently thus our better architecture may become technically available in the future.

Finally, we compare the impact of the employment of graph matrices in the values of the attention mechanism. We set $G_{v,} = 1$ and use one local and two global symmetric attention modules (Equation 6) with rotation invariance, see Figure 12. The graph matrix $G_{v,}$ leads to a decrease in generalisation performance on goal change tasks and an increase in performance on the difficult environment involving many lava-crossings. Conceptually, the graph-matrix $G_{v,}$ admits some similarities to convolutions, which are known to preform better on limited sample size (Hassani et al., 2022). Thus it may be easier for this architecture to identify two-dimensional information while setting it to the identity restores some degree of permutation invariance.

**Table 4:** We experiment on the Lavacrossing Minigrid environment.

| Domain Train | Test Task | CNN | ViT | PI-ViT | SiT | SiT-1 |
|---|---|---|---|---|---|---|
| | *train: easy-N1* | **0.96 ± 0.01** | 0.23 ± 0.41 | 0.01 ± 0.0 | **0.94 ± 13.0** | 0.90 ± 0.23 |
| | *train: medium-N2* | **0.69 ± 0.43** | 0.25 ± 0.42 | 0.01 ± 0.0 | **0.84 ± 0.32** | 0.70 ± 0.42 |
| **Lavacrossing N1 & N2** | hard-N3 | **0.54 ± 0.47** | 0.19 ± 0.38 | 0.01 ± 0.0 | **0.46 ± 0.48** | 0.41 ± 0.47 |
| **medium &easy** | rotations average | 0.00 ±0.00 | 0.15 ± 0.34 | 0.01 ± 0.0 | **0.91 ± 0.21** | **0.91 ± 0.22** |
| | goal top-right | 0.08 ±0.27 | 0.21 ± 0.40 | 0.01 ± 0.0 | **0.71 ± 0.44** | **0.90 ± 0.24** |
| | goal bottom-left | 0.08 ±0.25 | 0.20 ± 0.40 | 0.01 ± 0.0 | **0.71 ± 0.44** | **0.90 ± 0.24** |
| | random goal | 0.07 ±0.25 | 0.15 ± 0.35 | 0.01 ± 0.0 | **0.24 ±42.0** | **0.21 ± 0.40** |

**Table 5:** Ablations study SiT for different number of local and global layers as well as using the symmetric part of SiT$^*$, defined in the appendix.

| Train | Test Task | ViT | | SiT | | | |
|---|---|---|---|---|---|---|---|
| | | 2 global | 1 loc./2 glob. | symmetric 1/1 | 1 loc./1 glob. | 1 loc./2 glob. | 2 global |
| | *train: easy-N1* | 0.08 ± 0.27 | 0.23 ± 0.41 | 0.86 ± 0.29 | 0.90 ± 0.23 | **0.93 ± 18.0** | 0.87 ± 0.28 |
| | *train: medium-N2* | 0.12 ± 0.31 | 0.25 ± 0.42 | 0.88 ± 0.26 | 0.70 ± 0.42 | 0.84 ±0.32 | 0.59 ± 0.46 |
| **Lavacr.** | hard-N3 | 0.10 ± 0.29 | 0.19 ± 0.38 | 0.45 ± 0.43 | 0.41 ± 0.47 | 0.46 ±0.48 | 0.03 ± 0.16 |
| **N1/N2** | rotations avg. | 0.03 ± 0.16 | 0.15 ± 0.34 | 0.88 ± 0.27 | **0.91 ± 0.22** | **0.95 ± 0.1** | 0.91 ± 0.21 |
| | goal top-right | 0.03 ± 0.15 | 0.21 ± 0.40 | 0.81 ± 0.35 | **0.90 ± 0.24** | **0.71 ± 0.44** | 0.60 ± 0.48 |
| | goal bottom-left | 0.02 ± 0.13 | 0.20 ± 0.40 | 0.81 ± 0.35 | **0.90 ± 0.24** | **0.71 ± 0.44** | 0.60 ± 0.48 |
| | random goal | 0.08 ± 0.27 | 0.15 ± 0.35 | **0.26 ± 0.43** | **0.21 ± 0.40** | **0.28 ± 0.44** | 0.12 ± 0.32 |

## G.2 MINIGRID - LAVACROSSING

In this section we briefly provide the details with error bars to the experiment provided in the main text. The tables 4, 5 and 6 contain the mean rewards averaged over 200 test episodes after 20M time-steps.

**Table 6:** Ablations study SiT for different number of local and global layers as well as using the symmetric part of SiT*, defined in the appendix. We set $G_{v,} = 1$ and use one local and one (as well as two) global symmetric attention modules equation 6 with rotation invariance.

| Train | Test Task | SiT | | SiT - $G_{v,} = 1$ | |
|---|---|---|---|---|---|
| | | 1 loc./1 glob. | 1 loc./2 glob. | 1 loc./1 glob. | 1 loc./2 glob. |
| | *train: easy-N1* | $0.90 \pm 0.23$ | $0.94 \pm 13.0$ | $0.92 \pm 0.19$ | $0.90 \pm 0.20$ |
| | *train: medium-N2* | $0.70 \pm 0.42$ | $0.84 \pm 0.32$ | $0.74 \pm 0.39$ | $0.59 \pm 0.46$ |
| **Lavacr. N1/N2** | hard-N3 | $0.41 \pm 0.47$ | $0.46 \pm 0.48$ | $0.42 \pm 0.47$ | $0.37 \pm 0.47$ |
| | rotations avg. | $0.91 \pm 0.22$ | $0.95 \pm 0.1$ | $0.94 \pm 0.13$ | $0.92 \pm 0.20$ |
| | goal top-right | $0.90 \pm 0.24$ | $0.71 \pm 0.44$ | $0.90 \pm 0.23$ | $0.86 \pm 0.29$ |
| | goal bottom-left | $0.90 \pm 0.24$ | $0.71 \pm 0.44$ | $0.90 \pm 0.23$ | $0.84 \pm 0.31$ |
| | random goal | $0.21 \pm 0.40$ | $0.28 \pm 0.44$ | $0.24 \pm 0.42$ | $0.29 \pm 0.44$ |

