# OpenReview forum: "SiT:   Symmetry-invariant Transformers for Generalisation in Reinforcement Learning"
_ICLR.cc/2024/Conference — Submitted to ICLR 2024_

### Official Review · Reviewer_Rs9f · 2023-10-27

**Soundness:** 3 good
**Presentation:** 2 fair
**Contribution:** 2 fair
**Rating:** 3
**Confidence:** 3

**Summary:**

This paper proposes a symmetry-invariant transformer (SiT) that learns invariant and equivariant latent representations on images, and applies it in some RL tasks. I think this paper fits better in the CV domain rather than RL, as most of the paper focuses on describing a symmetry-preserving ViT and offers little new insight for RL. Although leveraging symmetry is definitely helpful for promoting generalization, the paper does provide sufficient evidence on the benefits of using the proposed method in solving general RL tasks.

**Strengths:**

- Leveraging symmetries/invariances is a reasonable direction to improve the generalization performance in RL tasks.
- The paper proposes a method to enforce image-level symmetry-invariant/equivariant on ViT.
- The proposed method shows good performance on some vision-based RL tasks that require image-level generalization capability.

**Weaknesses:**

- The majority of the paper is to derive a symmetry-invariant and equivariant ViT model, and there is not much design in the RL part. That's why I think the paper should be treated as a CV paper rather than an RL paper. In that sense, the proposed method should at least first demonstrate its superiority in CV tasks. Unfortunately, the proposed method is only evaluated on the extremely simple CIFAR-10 dataset, and compared with no other CV baselines except for ViT.
- The proposed method is based on ViT, which makes its applicability only restricted to vision-based RL tasks. Moreover, as ViT is quite heavy and costly to learn, it inevitably hurts sample efficiency and usability as compared to other commonly used vision encoders in vision-based RL tasks. No learning curves or results are provided related to the sample efficiency for RL tasks, which is somewhat insufficient to demonstrate the practical value of the proposed method to the RL community.
- The evaluations are only conducted in tasks that particularly rely on image generalization capability, like Procgen and MiniGrid. For some tasks like robotic manipulation, as the sense of orientation is vital, the importance of symmetry-invariant may not be that large.
- There lack of strong baselines in the experiments. The baselines only include ViT, variants of the proposed method, and simple baselines like CNN. As far as I know, there are also some existing works that introduce symmetry-preserving designs in CNN or other image-based models. Such methods should be also compared given their relevance to this paper.
- Minor: the paper reminds me of a recent NeurIPS paper [1] that also uses symmetry to enhance RL performance. This is probably not super related as it enforces symmetry on the temporal aspect, but also worth mentioning.

[1] Cheng, P. Look Beneath the Surface: Exploiting Fundamental Symmetry for Sample-Efficient Offline RL. arXiv preprint arXiv:2306.04220.

**Questions:**

- How will the proposed method perform on more general visual control tasks, like Atari?

---

> ### Author Response · Authors · 2023-11-17
> **Author Response: Ran E2CNN baseline, Added Learning Curves,**
>
> **...inevitably hurts sample efficiency and usability as compared to other commonly used vision encoders in vision-based RL tasks. No learning curves or results are provided related to the sample efficiency for RL tasks, which is somewhat insufficient to demonstrate the practical value of the proposed method to the RL community.**
>
>
> - Indeed, we agree with the reviewer that sample efficiency curves provide a lot of insight. All SiT variants demonstrate comparable sample efficiency to the CNN baseline,  **see Figure 7a**.
> We have incorporated the training and testing curves for the main Procgen experiments into the rebuttal version. Transformers have enabled great advances in nlp and vision, particularly for scaling and our work possibly  opens these avenues for image-based RL.
>
>
> **symmetry-preserving designs in CNN or other image-based models. Such methods should be also compared**
>
> - We have added the E2CNN baseline and find SiTs outperform this baseline. We emphasize that aside from our performance gain we are the first to succeed in training  ViTs (SiTs) in RL on Procgen.
>
> **How will the proposed method perform on more general visual control tasks, like Atari?**
>
> - Atari doesn't allow for studying generalization easily, which is the focus of this work while procgen is widely used given its wide variety of levels, and a  more complex action-space, see e.g. [muZero].
>
> ---
> ---
>
>
> **1. ...the paper should be treated as a CV paper rather than an RL paper…**
>
> - **Relevance of Model Evaluation in RL**: Many studies in Reinforcement Learning (RL) focus on model architecture evaluation, including works directly relevant to ours such as those by Wang et al., and Ha et al. Other reviewers have also mentioned more examples in this context.
>
> **2. ...For some tasks like robotic manipulation, as… the importance of symmetry-invariant may not be that large…**
>
> - Actually,  invariant and equivariant networks [Wangx2] have been shown to give benefits in robotic control.
>
> **3. ...ViT is quite heavy and costly to learn, it inevitably hurts sample efficiency and usability as compared to other commonly used vision encoders in vision-based RL tasks…**
>
> - **(a) Transformers and ViTs in Vision-Based RL**: We agree, Transformers and Vision Transformers (ViTs) are more computationally intensive and less sample efficient, which has limited their application in vision-based RL. **This challenge was a primary motivation for our research. We believe our approach shows promise in altering this landscape.**
>
> - **(b)** Given the dominance of ViTs in Vision and Transformers in NLP, it's plausible that improvements in Transformer technology will similarly revolutionize vision-based RL, with ViTs becoming predominant. Recent technical advancements, such as efficient Transformers [flashattn], which offer up to a 10x performance boost may lead the way.
>
> **…The proposed method is based on ViT, which makes its applicability only restricted to vision-based RL tasks…**
>
> -  **(a)**  Vision-based RL is a pivotal subfield of deep RL, marking the origin and significant advancements in the field. The focus of our paper on this area upholds the relevance and depth of our research.
>
> - **(b)** **Other Application of Transformers in RL**: Moreover, transformers are central to Decision Transformers [a] and the recent IRIS [b]. Our approach is also applicable to these models. We originally chose not to elaborate further on 1D-data to not overload the paper conceptually. **However, it would be a minor change  to add a paragraph on 1D-data of SiTs. see appendix A.1 where we have added such a brief discussion.**
>
> ---
> ---
>
> **Minor: the paper reminds me of a recent NeurIPS paper [1] that also uses symmetry to enhance RL performance.**
>
> - We have added the suggested reference, even though we concur that it is somewhat orthogonal to our paper.
>
>
> ---
> References
> ---
> [muzero] Procedural Generalization by Planning with Self-Supervised World Models, A.Anand, J. Walker, Yazhe Li, Eszter Vértes, Julian Schrittwieser, Sherjil Ozair, Théophane Weber, Jessica B. Hamrick :https://arxiv.org/abs/2111.01587
>
> [Wangx2]
> Dian Wang and Robin Walters. So (2) equivariant reinforcement learning. In International Conference on Learning Representations, 2022
> Rui Wang, Robin Walters, and Rose Yu. Approximately equivariant networks for imperfectly symmetric dynamics. PMLR, 2022b.
>
> [flashattn] FlashAttention: Fast and Memory-Efficient Exact Attention with IO-Awareness Tri Dao, Daniel Y. Fu, Stefano Ermon, Atri Rudra, Christopher Ré https://arxiv.org/abs/2205.14135

---

> > ### Comment · Reviewer_Rs9f · 2023-11-22
> > **Thanks for the response**
> >
> > I've read the response from the authors. I still hold a generally negative impression of this paper.
> >
> > I think the primary weakness of the current paper is the lack of comprehensive evaluation. As I said, most of the designs proposed by this paper are on the CV side rather than RL, hence the paper should at least first demonstrate its superiority against related CV models. From the RL perspective, the community typically desires a vision-encoder module to be broadly applicable to a wide range of tasks and can be relatively lightweight to avoid hurting sample efficiency too much. That's why I think adding Atari tasks should be helpful to improve the paper. If the method can only provide improvements on special tasks that heavily depend on image generalization capabilities, like Procgen and MiniGrid, it might not be that attractive for the general RL community, especially since the proposed model is somewhat heavy.
> >
> > I think the paper can be greatly improved by adding more representative baselines, experimental evaluations on more CV tasks and other general vision-based RL tasks. All these will make the paper more convincing and fully demonstrate its usefulness for the RL community. But I don't think the paper is ready in its current shape.

---

> > > ### Author Response · Authors · 2023-11-22
> > > **Points  in our response not addressed**
> > >
> > > **Atari:**  We appreciate your suggestion to include Atari tasks for a more comprehensive evaluation.   Could you please elaborate on how Atari tasks might provide a different measure of generality compared to Procgen? Procgen and MiniGrid as they are widely recognized benchmarks for testing generalization in vision-based RL tasks. Note that Fruitbot and Starpilot do not rely on symmetries, so they are not specifically chosen but probably the most discussed Procgen environments. Note that we do have a proof of concept on DM-control walker-walk in the paper. )
> > >
> > > **CV focus:** We would be grateful if you could further elucidate your viewpoint, particularly how our contribution contrasts to other RL paper doing model evaluation  (several cited in our work & published as RL paper in major conferences )?

---

> > > > ### Author Response · Authors · 2023-11-23
> > > > **Atari 100k**
> > > >
> > > > **We have incorporated four Atari evaluations as per your request. We hope this addition addresses your concerns effectively. Thank you for advocating for this inclusion; we believe it significantly enhances the value of our paper.**

---

### Official Review · Reviewer_vdZg · 2023-10-30

**Soundness:** 3 good
**Presentation:** 3 good
**Contribution:** 3 good
**Rating:** 8
**Confidence:** 4

**Summary:**

This paper introduces Symmetry-Invariant Transformer (SiT), a self-attention based network architecture that incorporates planar rotation and reflection symmetries. Central to the architecture is the proposed Graph Symmetric Attention (GSA) layer, which utilizes a graph topology matrix $G$ to control the different symmetries that are allowed in the layer by breaking the full-permutation invariance in the standard attention mechanism. By employing GSA both locally (i.e., each pixel is a token) and globally (i.e., each image patch is a token), SiT efficiently encodes symmetries at various levels. Moreover, by changing the token embedding from invariant features to equivariant features, SiT can be extended to be equivariant (or both invariant and equivariant). The authors apply SiT in both reinforcement learning and supervised learning, showing a solid improvement in both performance and sample efficiency.

**Strengths:**

1. The proposed architecture is novel. It provides a fresh perspective that connects permutation equivariance and rotation/reflection equivariance by constraining the grid topology matrix.
2. Utilizing GSA both at local and global levels to preserve symmetries across varying scales is an appealing concept.

**Weaknesses:**

1. The experiments could benefit from stronger baselines. Given the paper's introduction of a novel equivariant architecture, I think a comparison with existing equivariant architectures is necessary. A possible baseline could be some equivariant architectures that enforce global rotational symmetries like e2cnn[A], or utilizing rotation data augmentation to realize equivariance in CNN or ViT.
2. It would be nice to have an ablation study on only using GSA globally or locally to understand how the two components contribute to the improvement.

[A] Weiler, Maurice, and Gabriele Cesa. "General e (2)-equivariant steerable cnns." Advances in neural information processing systems 32 (2019).

**Questions:**

1. In the first paragraph of page 4, the paper claims, `Notably, GSA reduces to the conventional attention mechanism when the underlying graph topology matrix G only contains self-loops, i.e. G being the identity matrix.` My interpretation of this is that if $G$ is the identity matrix, the output of $\Gamma(Q, K)$ would result in a diagonal matrix, where each token can only attend to itself. This doesn't align with the conventional attention mechanism. I think $G$ should be an all-one matrix here instead of an identity matrix. Please correct me if I am missing something here.
2. I do not fully understand why $G_{k,v,q}$ are necessary. Would just having $G$ in equation 3 not be enough for maintaining the desired equivariance?
3. How are the sizes of the local patch (5x5) and the global patch (14x14) selected?

---

> ### Author Response · Authors · 2023-11-17
> **Author response: Added E2CNN baselines, Ablation Results, Clarifications**
>
> We thank the reviewer for taking the time to provide their helpful and valuable feedback.
> Our response follows:
>
> **The experiments could benefit from stronger baselines. …. like e2cnn.**
>
> - We have added the E2CNN baseline, which outperforms CNNs on Cavelfyer and Chaser, and find that SiTs outperform this stronger baseline. We emphasize that aside from our performance gain, we are the first to succeed in training transformer-based models (SiTs) on Procgen RL tasks.  Transformers have enabled great advances in NLP and vision, particularly for scaling and our work possibly opens these avenues for image-based RL.
>
> **it would be nice to have an ablation study on only using GSA globally or locally…**
>
> - On Minigrid Lavacrossing, SiT with 2 global GSA layers falls short compared to 1 local & 1 global and 1 local & 2 global layers, respectively; in particular, on the most challenging generalization test tasks i.e. random goal and hard (N3) environment.  **For reference, please see the appendix section G1, particularly Figure 10 and Table 4, where we have explored this aspect.** However, isolating the effects of local GSA alone presents architectural design challenges. But the local effects can be inferred indirectly by comparing 1 loc./2 glob. to 2 glob. in  Figure 10 and Table 4.
>
>
>
> ---
> **Answers to Questions**
> ---
>
> **... I think G should be an all-one matrix here instead of an identity matrix…**
>
> - In order to get the conventional attention mechanism, G needs  to be the identity matrix. Please review Equations 17 and 18 in the appendix, where we detail the index contractions of GSA. Since we apply graph matrices for each feature, the identity matrix is replicated for the number of features $ d_f$. The intriguing observation that this corresponds to a graph without connections to other vertices aligns nicely with our graph interpretation of G.
>
> **I do not fully understand why  $ G_{v,k,q} $  are necessary…. Would just having  G  in equation 3 not be enough for maintaining the desired equivariance?**
>
> - Correct, both  $G$ and $ G_{v,k,q}$ are indeed sufficient to break symmetries, respectively, i.e. to achieve the desired equivariance. We delve into this more in the paragraph titled *Explicit & Adaptive Symmetry Breaking* on page 5.  In our preliminary experiments, G required more h-param tuning- probably due to the Hadamard product -, thus we employ  $ G_{v,k,q} $. We have added this discussion to the paper.
>
>
> **How are the sizes of the local patch (5x5) and the global patch (14x14) selected?**
>
> - For Minigrid, we selected a 14x14 global image size as this is the minimum resolution at which the direction of the triangular shaped agent can be discerned when downscaling the RGB rendering of the environment. Smaller resolutions fail to capture this detail, while larger ones are feasible but not necessary. The local neighborhood size in Minigrid (in pixels) is required to be odd, so 3x3 and 5x5 are the smallest viable options. We have revised the paper to include this detail.

---

> > ### Comment · Reviewer_vdZg · 2023-11-22
> >
> > I would like to thank the authors for their rebuttal and new experiments, all of my concerns are properly addressed, and I would like to increase my score to 8.

---

> > > ### Author Response · Authors · 2023-11-22
> > > **Thankful for Your Positive Feedback**
> > >
> > > We're grateful for your recognition of our improvements and for the improved score!

---

### Official Review · Reviewer_Yry2 · 2023-10-31

**Soundness:** 3 good
**Presentation:** 2 fair
**Contribution:** 3 good
**Rating:** 6
**Confidence:** 4

**Summary:**

This paper introduces Symmetry-Invariant Transformer (SiT), a variant of Vision Transformer that can identify and leverage local and global data patterns. SiT employs Graph Symmetric Attention to maintain graph symmetries, creating invariant and equivariant representations. SiT's contributions involve addressing generalization challenges in RL by utilizing self-attention to handle both local and global symmetries, resulting in better adaptation to out-of-distribution data distributions. It surpasses Vision Transformers (ViTs) and CNNs in RL benchmarks like MiniGrid and Procgen showing improved generalization ability. SiT's contributions include handling symmetries at the pixel level, achieving superior RL performance, and introducing novel methods for employing graph symmetries in self-attention without relying on positional embeddings.

**Strengths:**

1. This paper provides an interesting way to achieve equivariance to different symmetries in grid data by using graph symmetric attention and different choices of graph topology matrix $G$. To the best of my knowledge, this idea is novel.

2. SiT achieves impressive generalization performance on MiniGrid and ProcgenRL over ViTs and CNNs.

**Weaknesses:**

1. Some sections of this paper are not well written leading to confusion while following the paper's arguments. For example in equation 2, symmetric($GV$) assumes that GV is $\mathbb{R}^{P\times P}$ whereas according to the author's definition of $G$ it should be $\mathbb{R}^{P\times d_f}$. This makes it really hard to follow the author's argument and how this formulation is connected to Graph Attention Networks [1] or $E(n)$ Equivariant GNN [2], This also makes following section 3 difficult.

2. As SiT has been built keeping in mind the inductive biases coming from the symmetries of the environment and task, I think just comparing with CNN or ViT baseline is not completely fair. The authors should use an E2 equivariant architecture like E2 Steerable Networks as their baseline for RL experiments [3, 4] or E(n) equivariant GNN [2]. Authors should also expand on the related work on equivariant architectures [5,6] for reinforcement learning. [3, 4, 7]

[1] Petar Velicˇkovic ́, Guillem Cucurull, Arantxa Casanova, Adriana Romero, Pietro Liò, and Yoshua Bengio. Graph attention networks. In International Conference on Learning Representations, 2018

[2] Victor Garcia Satorras, Emiel Hoogeboom, and Max Welling. E(n)-equivariant graph neural networks. In International Conference on Learning Representations, 2021a

[3] Dian Wang and Robin Walters. So (2) equivariant reinforcement learning. In International Conference on Learning Representations, 2022

[4] Arnab Kumar Mondal, Pratheeksha Nair, Kaleem Siddiqi. Group Equivariant Deep Reinforcement Learning ICML 2020 Workshop on Inductive Biases, Invariances and Generalization in RL

[5] Taco S Cohen, Max Welling Group Equivariant Convolutional Networks Proceedings of the International Conference on Machine Learning 2016

[6] Maurice Weiler, Gabriele Cesa. General E (2)-Equivariant Steerable CNNs. Advances in Neural Information Processing Systems 2019

[7] Linfeng Zhao, Xupeng Zhu, Lingzhi Kong, Robin Walters, Lawson LS Wong. Integrating Symmetry into Differentiable Planning with Steerable Convolutions. ICLR 2023

**Questions:**

Q1. In Figure 1 (a), what does the bottom right image depict?


Q2. Can you explain how Equation 2 is connected to GAT and E2GNN?

---

> ### Author Response · Authors · 2023-11-17
> **Author Response: Ran E2CNN baseline, added citations, Clarifications**
>
> **...The authors should use an E2 equivariant architecture like E2 Steerable Networks as their baseline for RL experiments…**
> - We have added the E2CNN baseline, which outperforms CNNs on Caveflyer and Chaser, and find that SiTs outperform this stronger baseline. We emphasize that aside from our performance gain, we are the first to succeed in training transformer-based models ViTs (SiTs) on Procgen RL tasks. in RL on Procgen.   Transformers have enabled great advances in NLP and vision, particularly for scaling and our work possibly opens these avenues for image-based RL.
>
> **Some sections of this paper are not well written leading to confusion … for example in equation 2, symmetric(GV) assumes that G is  $ \mathbb{R}^{P \times P} $ whereas according to the author's definition of  G it should be  $ \mathbb{R}^{P \times d_f} $.**
>
> - We have made the background section more concise in the rebuttal version, in particular the discussion around equation 2, which may have caused this confusion. We define G as $G \in \mathbb{R}^{P \times P}$. See e.g., in the definition of G page 4, or Figure 2, which visualizes G . Also we could not locate any such typo in our manuscript.  We do apply the graph matrix per each feature so there is an additional $d_f$ in the dimensionality as  $G \in \mathbb{R}^{P \times P \times d_f}$.
>
> **...Authors should also expand on the related work on equivariant architectures [5,6] for reinforcement learning. [3, 4, 7]**
>
> - We have included the suggested references, and mentioned them in the related work section. While E2CNN networks abide by equivariance they are based on CNNs in contrast to our ViT based approach, which moreover allows for invariance, equivariance as well as a mixture thereof i.e. SieT.
>
> ---
> **Answers to Questions**
> ---
>
>
> **In Figure 1 (a), what does the bottom right image depict?**
>
> - Figure 1 (a) bottom-right shows the original image where a minimal number of patches are permuted (six to be precise). This highlights that even minimal permutation invariance of an agent may be fatal for learning - aligned with our motivation that PI generally should be broken by some mechanism i.e. GSA.  We have clarified this in the revision
>
> **Can you explain how Equation 2 is connected to GAT and E2GNN?**
>
> - Equation 2 is given to provide an overview formulation of GAT. There is no direct connection of Equation 2 to E2GNN (Satorras et al., 2021b). We have clarified this in the revision.
>
> **...This makes it really hard to follow the author's argument and how this formulation is connected to Graph Attention Network…**
>
> - It appears the above misunderstanding may have been caused by the appearance of “G” in  equation 2, which is not the  G introduced in our work e.g. in Figure 2. We have changed the notation to $\mathcal{G}$ in equation 2 for the adjacency matrix.

---

> > ### Comment · Reviewer_Yry2 · 2023-11-22
> > **Official Response to Authors**
> >
> > I want to thank the authors for their detailed response to my concerns and for adding new experiments with stronger baselines.
> >
> > I think some of reviewer Rs9f's concerns are valid. The positioning of this paper is a little tricky. If the paper is about Symmetry and RL then it might make sense to do a more comprehensive literature review of related work in RL which tries to tackle data-efficiency and generalization. For example, papers like [1,2] should be added in related work and it should be discussed how the current technique in comparison to those and argument about why they did not compare with those. I agree it might not make sense to compare with everything but putting the current paper into the landscape of symmetry RL work is necessary if that's the main area the authors are targeting. Moreover, if the authors have Atari 100k experiments they can compare with [2] and show how their method performs. I don't think they need to beat everything. But having it might be useful for a complete picture.
> >
> > For example, in this line in the Related Work section, "Equivariant neural networks and symmetric representation learning have contributed to sample-efficient RL (Laskin et al., 2020; Yarats et al., 2021a; van den Oord et al., 2018)." None of these papers are on symmetric representation learning. They are on SSL and data augmentation in RL to learn representations that are invariant to augmentations and not equivariant.  [1] and [2] are better references for symmetric representation learning for RL and are more relevant to this work.
> >
> > I agree with the authors that the paper doesn't necessarily need to be about equivariant architectures for images and can focus on RL if that's the problem the authors are targeting. However, I think the paper can be improved further in terms of its positioning in the Symmetry RL literature.
> >
> > I have increased the score to appreciate the author's effort in improving the clarity and including proper baselines.
> >
> >
> > [1]  Rezaei-Shoshtari, Sahand, et al. "Continuous MDP Homomorphisms and Homomorphic Policy Gradient." Advances in Neural Information Processing Systems 35 (2022): 20189-20204.
> >
> > [2]  Mondal, Arnab Kumar, et al. "Eqr: Equivariant representations for data-efficient reinforcement learning." International Conference on Machine Learning. PMLR, 2022.

---

> > > ### Author Response · Authors · 2023-11-23
> > > **Atari 100k**
> > >
> > > We're grateful for your acknowledgement of our improvements and thank you for the score increase! Please see the official comment to address your remaining points!

---

### Author Response · Authors · 2023-11-17

We thank the reviewers for their valuable feedback. Reviewers find the ideas presented to be novel and empirical results on RL generalization to be solid. Based on reviewers' comments, we made the following changes:

- Added **additional baseline using the E2CNN**, see below or Table 1 in the rebuttal submission. **[Reviewer Yry2, vdZg Rs9f]**
- All **SiT variants are comparable in sample efficiency to the CNN baseline**. Procgen training/testing curves have been added  for completeness,  see Figure 7a. **[Reviewer Rs9f]**

| Procgen Task | CNN   | E2CNN       |**SiT***  | **SieT**   |
|--------------|-------|-------------|-------|--------|
| CaveFlyer    | 4.0%  | 17.7% | 55.5% | 34.5%  |
| StarPilot    | 36.3% | 29.4% | 31.0% | 42.2%  |
| Fruitbot     | 70.8% | 66.1% | 70.5% | 76.0%  |
| Chaser       | 10.6% | 15.6% | 45.6% | 54.0%  |
| **Average**  | 30.4% | 32.2%| **50.6%** | **51.7%**  |

---

> ### Author Response · Authors · 2023-11-23
> **Atari 100k**
>
> We're grateful for your acknowledgement of our improvements and thank you for the score increase!
>
> **For example, papers like [1,2] should be added in related work and it should be discussed how the current technique in comparison to those and argument about why they did not compare with those.**
>
> We thank the reviewer for their suggestions. To address their main concern, we have updated the related work section to better position our work with respect to prior work on symmetry in RL.
>
> While algebraic symmetries in RL and symmetry-based representation learning  can improve sample efficiency in a single environment using CNN-based architectures as backbone; the focus of our work is on improving generalization in RL by incorporating symmetries in vision transformers. Due to the widespread use of transformers in NLP and vision, transformer-based architectures carry many additional promises for future development in RL, see our revised conclusion section for examples. Interestingly, SiTs are complimentary and may be added on top of some prior works incorporating symmetries e.g. [1,2], which we leave for future work.
>
> **if the authors have Atari 100k experiments they can compare with [2] and show how their method performs.**
>
> Based on the discussion with reviewer Rs9f,  we ran some Atari 100k experiments to
> serve as a proof-of-concept that SiTs can be extended easily to other tasks. That said, we want to reiterate that the focus of our work is on generalization in RL as opposed to data efficiency.
>
> Specifically, we train SiT on top of Rainbow **without h-parm changes**, and find that SiTs compare similarly to CNNS, see Table in section F.3. To the best of our knowledge, this is the first time ViTs + Q-learning  has been successfully applied for online RL on Atari.

---

### Comment · Area_Chair_o8i4 · 2023-11-21
**Reviewers: Please respond to authors or update review**

Dear Reviewers,

The discussion phase will end tomorrow.  Could you kindly respond to the authors rebuttal letting them know if they have addressed your concerns  and update your review as appropriate? Thank you.

-AC

---

### Meta-Review · Area_Chair_o8i4 · 2023-12-11

**Metareview:**

**Summary** This paper proposes a new model Symmetry-Invariant Transformer (SiT), a self-attention based architecture.  SiT is designed to incorporate local and global symmetries in order to improve generalization and sample efficiency.  A key innovation is the Graph Symmetric Attention (GSA) layer which breaks the full permutation symmetry of self-attention to that of a graph.  To encode symmetry at different levels, GSA is applied locally (pixel-wise) and globally (patch-wise).  SiT can adapt to both equivariant or invariant features or a combination.  The model is evaluated on RL benchmarks MiniGrid, Procgen, Atari 100K, and DM Control and on vision benchmark CIFAR-10.

**Metareview** The presented method is novel and interesting.  It provides a fresh perspective on rotational equivariance.  In particular the incorporation of local and global symmetries is quite interesting and has demonstrated empirical value through beating E(2)-CNN in benchmarks.  Performance on Procgen is good, beating ViTs, CNNs, and E(2)-CNN.  The paper's weaknesses include presentation quality, which could be better.   Most critically, reviewers debated the sufficiency of the evaluation.  This is an RL method and thus vision benchmarks are not critical, however, they may strengthen the paper by further illuminating the strengths and weaknesses of the method.  There is also value in demonstrating applicability across a broad range of RL tasks.  To address the concern, the authors added an experiment on Atari 100K, however, performance was not good.  This method is quite innovative and has the potential for impact, but the issues with evaluation should be improved before publication.

**Justification For Why Not Higher Score:**

- need more thorough evaluation on more tasks to demonstrate model flexibility

**Justification For Why Not Lower Score:**

N/A

---

### Decision · Program_Chairs · 2024-01-16

Reject